# FEATURE-AWARE (HYPER)GRAPH GENERATION VIA NEXT-SCALE PREDICTION

## ABSTRACT

Graph generative models have shown strong results in molecular design but struggle to scale to large, complex structures. While hierarchical methods improve scalability, they usually ignore node and edge features, which are critical in real-world applications. This issue is amplified in hypergraphs, where hyperedges capture higher-order relationships among multiple nodes. Despite their importance in domains such as 3D geometry, molecular systems, and circuit design, existing generative models rarely support both hypergraphs and feature generation at scale. In this paper, we introduce FAHNES (**f**eature-**a**ware **h**ypergraph generation via **ne**xt-**s**cale prediction), a hierarchical framework that jointly generates hypergraph topology and features. FAHNES builds multi-scale representations through node coarsening and refines them via localized expansion, guided by a novel node budget mechanism that controls granularity and ensures consistency across scales. Experiments on synthetic, 3D mesh and graph point cloud datasets show that FAHNES achieves state-of-the-art performance in jointly generating features and structure, advancing scalable hypergraph and graph generation.

## 1 INTRODUCTION

Generating discrete geometric structures such as graphs and hypergraphs is an important challenge in modern machine learning (Zhu et al., 2022). These structures are central to applications ranging from molecular design and materials discovery to electronic circuits and 3D shape modeling (Kajino, 2019; Rahman et al., 2012; Starostin & Balashov, 2008; Luo et al., 2024). Their ability to capture complex relationships—pairwise in graphs, and multi-way in hypergraphs—makes them an essential tool for representing and synthesizing structured data. In many real-world settings, topology alone is not enough: node and edge (or hyperedge) features often carry important semantic or geometric information, such as atom and bond types in molecules, or vertex coordinates in 3D meshes.

Despite recent advances, existing methods for featured graph generation struggle to scale. Most of these approaches use flat architectures that model the entire structure at once, leading to quadratic computational and memory complexities (Vignac et al., 2023; Eijkelboom et al., 2024; Xu et al., 2024). This limits their use to moderately sized graphs. To address scalability, hierarchical generation strategies have been proposed for both graphs and hypergraphs (Bergmeister et al., 2024; Gailhard et al., 2025). These methods build the structure in stages, starting from a coarse representation and progressively upsampling and refining it, which allows them to handle much larger topologies.

However, existing hierarchical methods only focus on unfeatured structures (Bergmeister et al., 2024; Gailhard et al., 2025). Extending them to generate features is challenging because different regions of a graph or hypergraph often grow at uneven rates during the refinement process, making it difficult to maintain consistency across scales. Furthermore, sequentially generating topology first and then features—rather than modeling them jointly—is ineffective in complex settings, as illustrated in Figure 1.

To overcome these limitations, we propose FAHNES (**f**eature-**a**ware **h**ypergraph generation

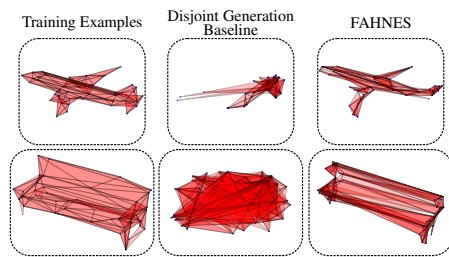

Figure 1: Examples of generated featured hypergraphs by a sequential disjoint generation baseline and our model (FAHNES).

via **ne**xt-**s**cale prediction), a hierarchical generative framework that jointly models topology and features for both graphs and hypergraphs[1]. FAHNES builds multi-scale representations (Tian et al., 2024) through node coarsening and reconstructs fine structures via localized expansion, while directly predicting features alongside structure at each stage. A fundamental component of our model is a *node budget mechanism*, which encodes local growth constraints and helps maintain consistency in regions that expand at different rates. In addition, we adapt minibatch *flow-matching optimal-transport* (OT) coupling (Tong et al., 2024; Pooladian et al., 2023) to the hierarchical setting, improving stability and coherence during generation. This combination enables scalable generation of large, featured graphs or hypergraphs across diverse domains, from 3D meshes to point clouds. Our main contributions are:

- We introduce the first scalable hierarchical model for the joint generation of hypergraph topology and features, targeting complex data such as 3D meshes and point clouds.
- We propose a novel budget-based mechanism that enhances global structural coherence in hierarchical generation (Sections 3.2 and 3.3).
- We extend minibatch OT-coupling to hierarchical graph or hypergraph generation, improving generative stability and quality (Section 3.5).
- We validate FAHNES on both synthetic and real-world datasets, demonstrating strong performance in jointly generating topology and features (Section 4).

## 2 RELATED WORK

**Graph and hypergraph generation using deep learning**. Graph generation has seen significant advances in recent years. Early approaches, such as GraphVAE (Simonovsky & Komodakis, 2018), employed autoencoders to embed graphs into latent spaces for sampling. Subsequent models leveraged recurrent neural networks to sequentially generate adjacency matrices, improving structural fidelity (You et al., 2018). More recently, diffusion-based methods have enabled permutation-invariant graph generation (Niu et al., 2020; Vignac et al., 2023), with extensions incorporating structural priors such as node degrees (Chen et al., 2023). Many of these previous methods jointly model node features and topology, but are limited to small graphs due to scalability challenges. Diffusion-based models (Vignac et al., 2023; Eijkelboom et al., 2024; Xu et al., 2024) operate over complete graphs by gradually corrupting structures and features, then training models to denoise them. However, their scalability is constrained by the combinatorial number of possible edges in graphs.

To mitigate this, hierarchical methods have been proposed. Bergmeister et al. (2024) introduced a scalable graph generation framework based on a coarsen-then-expand approach, merging nodes to form coarse representations and progressively reconstructing finer details. This framework was extended to hypergraphs by Gailhard et al. (2025), which allows edges to connect more than two nodes. However, both methods focus exclusively on topology generation, neglecting node and hyperedge features essential for many applications.

Hierarchical graph generation has also been explored extensively in molecular modeling. Coarse-to-fine diffusion frameworks (Qiang et al., 2023) generate coarse groups before refining their types and connectivity, but their two-stage design is difficult to extend beyond a small number of hierarchical levels and is highly specialized for molecules. Other hierarchical flow models such as MolGrow (Kuznetsov & Polykovskiy, 2021) and MolHF (Kuznetsov & Polykovskiy, 2021) rely on dequantization to backpropagate across discrete coarsening steps and use BFS-based merging rules tailored to molecular graphs. In contrast, our approach jointly models features and connectivity at all scales, uses a more general local-variation merging cost (Loukas, 2019), and is not restricted to molecular generation, enabling broader and more scalable hierarchical modeling.

**Applications of hypergraph generation**. Generative models that capture both higher-order structure and node/hyperedge features are critical in numerous domains. In molecular design, regular graph generative models struggle to accurately represent rings and scaffolds (Vignac et al., 2023; Barsbey et al., 2025), which naturally correspond to hyperedges involving multi-atom interactions rather than pairwise bonds. Similarly, 3D shape modeling involves surfaces such as triangles, quads, or general

---

[1]Since hypergraphs are a generalization of graphs, we present FAHNES on hypergraphs. The explanation for graphs follows straightforwardly.

polygons that extend beyond simple pairwise connectivity. Conventional approaches often rely on fixed topology, quantization, or autoregressive sequence modeling with transformers (Nash et al., 2020; Siddiqui et al., 2024), limiting their flexibility and scalability. Hypergraphs provide a more general framework by treating faces as hyperedges, enabling the joint generation of topology and features.

Unlike prior work that focuses solely on topology or relies on flat, non-scalable designs, we present the first unified, scalable approach for jointly modeling hypergraph structure and features. FAHNES enhances hierarchical generative modeling with feature integration, a novel node-budget mechanism, and flow-matching training tailored to our strategy.

# 3 FEATURE-AWARE (HYPER)GRAPH GENERATION VIA NEXT-SCALE PREDICTION

**Notations**. Throughout this paper, we use calligraphic letters, like $\mathcal{V}$, to represent sets, with their cardinality denoted by $|\mathcal{V}|$. Matrices are represented by bold uppercase letters (*e.g.*, $\mathbf{A}$), while vectors are indicated by bold lowercase letters (*e.g.*, $\mathbf{x}$). The transpose and point-wise multiplication operations are denoted by $(\cdot)^\top$ and $\odot$, respectively. $\lceil \cdot \rfloor$ denotes rounding to the nearest integer.

**Basic definitions**. We define a graph $G = (\mathcal{V}, \mathcal{E})$ as a pair consisting of a set of vertices $\mathcal{V}$ and a set of edges $\mathcal{E} \subseteq \mathcal{V} \times \mathcal{V}$. Graphs may also carry node and edge features, represented by matrices $\mathbf{F}_\mathcal{V} \in \mathbb{R}^{|\mathcal{V}| \times m}$ and $\mathbf{F}_\mathcal{E} \in \mathbb{R}^{|\mathcal{E}| \times l}$, where $m$ and $l$ denote the dimensionality of the features. Each edge $e \in \mathcal{E}$ corresponds to a pair $(u, v)$, indicating a connection between nodes $u$ and $v$. A bipartite graph $B = (\mathcal{V}_L, \mathcal{V}_R, \mathcal{E})$ is a special case of a graph where the vertex set is split into two disjoint subsets $\mathcal{V}_L$ and $\mathcal{V}_R$, and edges exist only between the two parts, *i.e.*, $\mathcal{E} \subseteq \mathcal{V}_L \times \mathcal{V}_R$. The full set of nodes is $\mathcal{V} = \mathcal{V}_L \cup \mathcal{V}_R$. In this work, we consider node features for bipartite graphs, denoted by $\mathbf{F}_L$ for the left-side nodes and $\mathbf{F}_R$ for the right-side nodes.

A hypergraph $H$ extends the concept of a graph and is specified by a pair $(\mathcal{V}, \mathcal{E})$, where $\mathcal{V}$ represents the vertex set and $\mathcal{E}$ comprises hyperedges, with each $e \in \mathcal{E}$ being a subset of $\mathcal{V}$. The main distinction of hypergraphs lies in their ability to connect arbitrary numbers of vertices through a single hyperedge. Similar to graphs, hypergraphs can possess node and hyperedge features, which are denoted $\mathbf{F}_\mathcal{V}$ and $\mathbf{F}_\mathcal{E}$. We consider two fundamental graph-based representations of hypergraphs: clique and star expansions. The clique expansion transforms a hypergraph $H$ into a graph $C = (\mathcal{V}, \mathcal{E}_c)$, where hyperedges are replaced by cliques, *i.e.*, $\mathcal{E}_c = \{(u, v) \mid \exists\, e \in \mathcal{E} : u, v \in e\}$. The star expansion converts a hypergraph $H$ into a bipartite structure $B = (\mathcal{V}_L, \mathcal{V}_R, \mathcal{E}_b)$, where $\mathcal{V}_L = \mathcal{V}$, $\mathcal{V}_R = \mathcal{E}$, and $\mathcal{E}_b = \{(v, e) \mid v \in \mathcal{V}_L, e \in \mathcal{V}_R, v \in e \text{ in } H\}$. Left side nodes $\mathcal{V}_L$ represent the nodes of the hypergraph, while right side nodes $\mathcal{V}_R$ represent hyperedges.

Our objective is to develop a generative model that can sample from the underlying distribution of a dataset of featured hypergraphs (or graphs) $(H_1, \ldots, H_N)$, *i.e.*, to learn the joint distribution over both topology and features. We present FAHNES in the more general setting of hypergraphs; since graphs are a special case of hypergraphs, the model can be directly adapted to graphs, as discussed in Section 3.6. Complete mathematical proofs of all propositions in this paper appear in Appendix A.

## 3.1 OVERVIEW

FAHNES follows a hierarchical pipeline as shown in Figure 2: *i*) during training, multi-scale representations of input (hyper)graphs are produced; *ii*) a model learns to reconstruct a higher scale from a lower one. Our approach adopts a coarsening–expansion framework (Gailhard et al., 2025). We first perform standard spectrum-preserving coarsening (Loukas, 2019) on the clique expansion of the hypergraph to obtain the multi-scale representation. The process is then reversed through expansion and refinement, reconstructing both topology and features at progressively finer scales. Expansion and refinement are learned on the bipartite representation using the flow-matching framework (Lipman et al., 2022), treating generation as the inverse of the coarsening process.

To address uneven growth across regions, we introduce a *node budget* mechanism: each cluster is assigned a budget indicating its remaining expansions. Budgets are recursively divided among child nodes, starting from a single super-node with the full budget and ending when all clusters have a budget of one. This provides more local control over the final node count and improves global

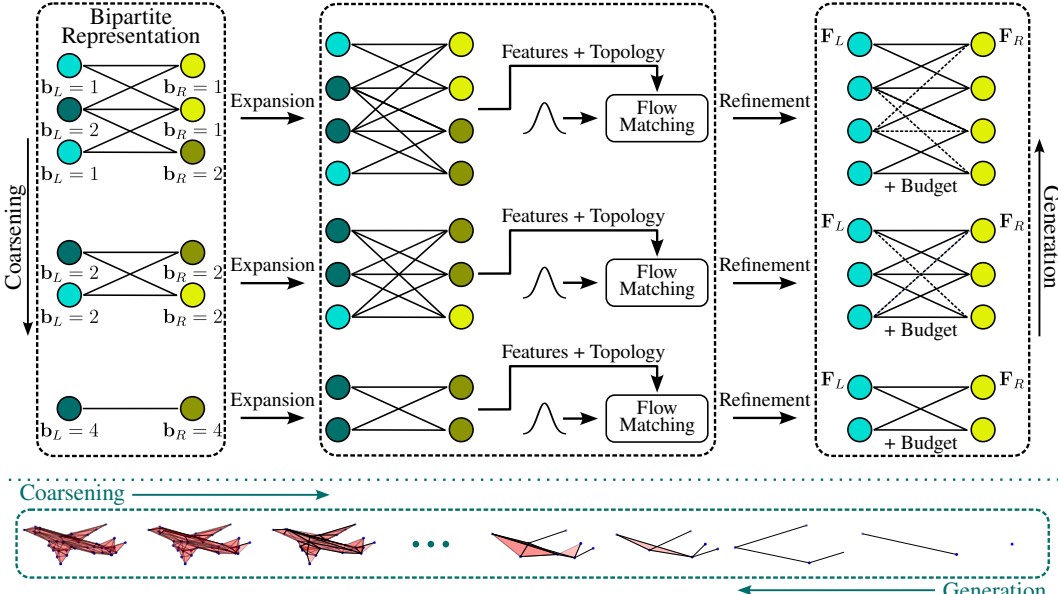

Figure 2: Our framework adopts a *coarsening-expansion* strategy. *i*) During training, input hypergraphs are progressively coarsened by merging nodes and hyperedges, yielding a multiscale representation. Node features are averaged during merging, and budgets are summed. *ii*) The model is then trained to predict which nodes were merged at each scale. *iii*) In the *expansion* phase, merged nodes (shown in dark in the leftmost column) are expanded back (copies shown in dark), inheriting their parent's features, budget, and connectivity. In the *refinement* phase, the model is trained to (*a*) identify which edges should be removed (dotted lines), (*b*) predict how the parent's budget should be split across the children, and (*c*) refine the features of newly expanded nodes.

consistency compared to prior methods (Bergmeister et al., 2024; Gailhard et al., 2025), which only append the desired size to node embeddings. At each expansion step, the model also predicts the mean feature of future child nodes conditioned on the parent's feature, extending the scale-wise autoregressive idea of Ren et al. (2024) so that predictions at one scale guide those at the next.

## 3.2 BUDGETED COARSENING

We adopt the coarsening strategy of Gailhard et al. (2025), representing the hypergraph at multiple resolution levels by merging nodes into clusters. Node pairs to be merged are selected using the spectrum-preserving coarsening of Loukas (2019), applied to the hypergraph's clique expansion. Mergings in the clique expansion are then mirrored in the bipartite representation, with corresponding left side nodes being merged, and copies of the same hyperedge being subsequently collapsed. We limit node mergings to at most two per cluster, which consequently bounds hyperedge mergings to at most three (Gailhard et al., 2025). For each resulting cluster, we track two quantities. First, the *budget* is the total number of nodes contained in the cluster. Initially, each node and hyperedge has a budget of 1; when clusters merge, their budgets are summed and assigned to the new super-cluster. The node budget provides a local growth constraint, indicating how many fine-level nodes each coarse cluster should expand into and guiding refinement in regions that grow at different rates. Second, the super-cluster's feature is computed as the weighted mean of the features of the clusters it absorbs, so it represents the average feature of all its contained nodes.

We denote node (left-side) budgets and features by $\mathbf{b}_L \in \mathbb{N}^{|\mathcal{V}_L|}$ and $\mathbf{F}_L$, and hyperedge (right-side) budgets and features by $\mathbf{b}_R \in \mathbb{N}^{|\mathcal{V}_R|}$ and $\mathbf{F}_R$. The following definition formalizes our coarsening process.

**Definition 1** (Bipartite graph coarsening). Let $H$ be an arbitrary hypergraph, $C = (\mathcal{V}_c, \mathcal{E}_c)$ its clique expansion, and $B = (\mathcal{V}_L, \mathcal{V}_R, \mathcal{E}_b, \mathbf{b}_L, \mathbf{b}_R, \mathbf{F}_L, \mathbf{F}_R)$ its *featured* bipartite representation. Let $\mathcal{P}_L =$

$\{\mathcal{V}^{(1)}, \ldots, \mathcal{V}^{(n)}\}$ be a partitioning[2] of the node set $\mathcal{V}_L$ such that each set $\mathcal{V}^{(p)}$ induces a connected subgraph in $C$. We construct an intermediate coarsening $\tilde{B}(B, \mathcal{P}_L) = (\bar{\mathcal{V}}_L, \mathcal{V}_R, \bar{\mathcal{E}}_b, \bar{\mathbf{b}}_L, \mathbf{b}_R, \bar{\mathbf{F}}_L, \mathbf{F}_R)$ by merging each part $\mathcal{V}^{(p)}$ into a single node $v^{(p)} \in \bar{\mathcal{V}}_L$, and by defining:

$$\bar{\mathbf{b}}_L[p] = \sum_{v \in \mathcal{V}^{(p)}} \mathbf{b}_L[v], \qquad \bar{\mathbf{F}}_L[p] = \frac{1}{\bar{\mathbf{b}}_L[p]} \sum_{v \in \mathcal{V}^{(p)}} \mathbf{b}_L[v]\mathbf{F}_L[v], \tag{1}$$

for every merged node $\mathcal{V}^{(p)}$. An edge $e_{\{p,q\}} \in \bar{\mathcal{E}}_b$ is added between $v^{(p)} \in \bar{\mathcal{V}}_L$ and $v^{(q)} \in \mathcal{V}_R$ if there exists an edge $e_{\{i,q\}} \in \mathcal{E}_b$ in the original bipartite representation between some $v^{(i)} \in \mathcal{V}^{(p)}$ and $v^{(q)}$.

To complete the coarsening process, we define an equivalence relation $v_1 \sim v_2 \iff \mathcal{N}(v_1) = \mathcal{N}(v_2)$ on $\mathcal{V}_R$, where $\mathcal{N}(v)$ denotes the set of neighbors of $v$, *i.e.*, we consider right side nodes having the same set of neighbors as equivalent, or in other words we consider hyperedges containing the same set of nodes as equivalent. This induces a partitioning $\mathcal{P}_R = \{\mathcal{V}_R^{(1)}, \ldots, \mathcal{V}_R^{(m)}\}$, allowing us to construct the fully coarsened bipartite representation $\bar{B}(\tilde{B}, \mathcal{P}_L) = (\bar{\mathcal{V}}_L, \bar{\mathcal{V}}_R, \bar{\mathcal{E}}_b, \bar{\mathbf{b}}_L, \bar{\mathbf{b}}_R, \bar{\mathbf{F}}_L, \bar{\mathbf{F}}_R)$ by merging each part $\mathcal{V}_R^{(p)}$ into a single node $v_R^{(p)} \in \bar{\mathcal{V}}_R$, similarly to the construction of $\bar{\mathcal{V}}_L$, and by defining:

$$\bar{\mathbf{b}}_R[p] = \sum_{v \in \mathcal{V}_R^{(p)}} \mathbf{b}_R[v], \qquad \bar{\mathbf{F}}_R[p] = \frac{1}{\bar{\mathbf{b}}_R[p]} \sum_{v \in \mathcal{V}_R^{(p)}} \mathbf{b}_R[v]\mathbf{F}_R[v], \tag{2}$$

for every merged hyperedge $\mathcal{V}_R^{(p)}$.

*Remark* 1. Informally, we cluster nodes in the clique expansion, merge the corresponding left-side nodes of the bipartite graph, and compute their budgets and features as described above. Right-side nodes (hyperedges) connected to the same left-side nodes are then merged. In our implementation, we select nodes for clustering using spectrum-preserving coarsening (Loukas, 2019). In the final reduction—when only one node and one hyperedge remain—we replace their features with zero matrices to yield a feature-agnostic initialization for generation. Please note that cluster size vectors introduced in (Bergmeister et al., 2024) correspond to the sizes of all sets $\mathcal{V}_L^{(p)}$ and $\mathcal{V}_R^{(p)}$, *i.e.*, the number of mergings for each cluster at this specific step, and not the number of nodes absorbed since the initialization of coarsening—which we call *budget*. Further details about the coarsening methodology are provided in Appendix B. Some visual examples of coarsening sequences are provided in Appendix C.

We provide the following theoretical result regarding the feature merging process in FAHNES.

**Proposition 1.** *Setting cluster features as the mean of the nodes they contain minimizes the mean squared error between the features of the fully expanded hypergraph and those of the original hypergraph.*

## 3.3 BUDGETED EXPANSION AND REFINEMENT

Once multi-scale representations for the dataset of hypergraphs are produced by the coarsening procedure, the objective is to train a model able to reverse it. In this section, we define the inverse of the coarsening procedure, which the model will learn to imitate. It comprises two stages: expansion ("upsampling" by duplicating nodes and hyperedges) and refinement (adjusting connectivity, budgets, and features). This process exclusively works on the bipartite representation.

Since our framework only conditions on the total number of nodes in the final hypergraph, we maintain a single node budget vector $\mathbf{b} \in \mathbb{N}^{|\mathcal{V}_L|}$ for the node (left-side) partition and discard the hyperedge (right-side) budget. To undo mergings, clusters are recursively split into multiple child nodes. Each child initially copies the parent's connections and inherits its feature and budget, before a subsequent refinement step, then: *i*) redistributes the budget among the children, *ii*) updates their features, and *iii*) removes edges that should not persist at the finer scale.

Formally, the expansion and refinement steps can be described as follows.

**Definition 2** (Bipartite graph expansion). Given a bipartite graph $B = (\mathcal{V}_L, \mathcal{V}_R, \mathcal{E}, \mathbf{b}, \mathbf{F}_L, \mathbf{F}_R)$ and two expansion size vectors $\mathbf{v}_L \in \mathbb{N}^{|\mathcal{V}_L|}$, $\mathbf{v}_R \in \mathbb{N}^{|\mathcal{V}_R|}$, denoting the number of duplication for nodes (left side) and hyperedges (right side), respectively. Let $\tilde{B}(B, \mathbf{v}_L, \mathbf{v}_R) =$

---

[2]That is, $\mathcal{V}^{(p)} \subseteq \mathcal{V}_L$, $\bigcup_{i=1}^{n} \mathcal{V}^{(i)} = \mathcal{V}_L$, and $\mathcal{V}^{(i)} \cap \mathcal{V}^{(j)} = \emptyset \ \forall \ 1 \le i, j \le n$.

$(\tilde{\mathcal{V}}_L, \tilde{\mathcal{V}}_R, \tilde{\mathcal{E}}, \mathbf{b}^{\text{expanded}}, \mathbf{F}_L^{\text{expanded}}, \mathbf{F}_R^{\text{expanded}})$ denote the expansion of $B$, whose node sets, budgets, and features are given by:

$$\tilde{\mathcal{V}}_L = \mathcal{V}_L^{(1)} \cup \cdots \cup \mathcal{V}_L^{(|\mathcal{V}_L|)}, \text{ where } \mathcal{V}_L^{(p)} = \{v_L^{(p,i)} \mid 1 \le i \le \mathbf{v}_L[p]\} \text{ for } 1 \le p \le |\mathcal{V}_L|,$$

$$\tilde{\mathcal{V}}_R = \mathcal{V}_R^{(1)} \cup \cdots \cup \mathcal{V}_R^{(|\mathcal{V}_R|)}, \text{ where } \mathcal{V}_R^{(p)} = \{v_R^{(p,i)} \mid 1 \le i \le \mathbf{v}_R[p]\} \text{ for } 1 \le p \le |\mathcal{V}_R|,$$

$$\mathbf{b}^{\text{expanded}}[p, i] = \mathbf{b}[p] \text{ for } 1 \le i \le \mathbf{v}_L[p], 1 \le p \le |\mathcal{V}_L|, \tag{3}$$

$$\mathbf{F}_L^{\text{expanded}}[p, i] = \mathbf{F}_L[p] \text{ for } 1 \le i \le \mathbf{v}_L[p], 1 \le p \le |\mathcal{V}_L|,$$

$$\mathbf{F}_R^{\text{expanded}}[p, i] = \mathbf{F}_R[p] \text{ for } 1 \le i \le \mathbf{v}_R[p], 1 \le p \le |\mathcal{V}_R|.$$

The edge set $\tilde{\mathcal{E}}$ includes all the cluster interconnecting edges: $\{e_{\{p,i;q,j\}} \mid e_{\{p,q\}} \in \mathcal{E}, v_L^{(p,i)} \in \mathcal{V}_L^{(p)}, v_R^{(q,j)} \in \mathcal{V}_R^{(q)}\}$.

*Remark* 2. Expansion thus acts as a *clone-and-rewire* operation: vertices and hyperedges are duplicated, and each child inherits every incident edge of its parent.

**Definition 3** (Bipartite graph refinement). Given a bipartite graph $\tilde{B} = (\tilde{\mathcal{V}}_L, \tilde{\mathcal{V}}_R, \tilde{\mathcal{E}}, \mathbf{b}, \mathbf{F}_L^{\text{expanded}}, \mathbf{F}_R^{\text{expanded}})$, an edge selection vector $\mathbf{e} \in \{0,1\}^{|\tilde{\mathcal{E}}|}$, a budget split vector $\mathbf{f} \in [0,1]^{|\tilde{\mathcal{V}}_L|}$, satisfying $\sum_{v \in \mathcal{V}_L^{(p)}} \mathbf{f}[v] = 1$ for all cluster $\mathcal{V}_L^{(p)}$ in $\tilde{\mathcal{V}}_L$, and two feature refinement vectors $\mathbf{F}_L^{\text{refine}}$ and $\mathbf{F}_R^{\text{refine}}$ with the same dimensions as $\mathbf{F}_L^{\text{expanded}}$ and $\mathbf{F}_R^{\text{expanded}}$, let $B(\tilde{B}, \mathbf{e}, \mathbf{f}, \mathbf{F}_L^{\text{refine}}, \mathbf{F}_R^{\text{refine}}) = (\tilde{\mathcal{V}}_L, \tilde{\mathcal{V}}_R, \mathcal{E}, \lceil \mathbf{b} \odot \mathbf{f} \rceil, \mathbf{F}_L^{\text{refine}}, \mathbf{F}_R^{\text{refine}})$ denote the refinement of $\tilde{B}$, where $\mathcal{E} \subseteq \tilde{\mathcal{E}}$ such that the $i$-th edge $e_{(i)} \in \mathcal{E}$ if and only if $\mathbf{e}[i] = 1$.

*Remark* 3. Edges are selectively removed based on the binary indicator vector $\mathbf{e}$, and features are updated with new predictions. Node budgets are divided among child nodes according to the split proportions specified by the vector $\mathbf{f}$. Since budgets must be integers, the resulting values are rounded. In the case of a tie (*e.g.*, when an odd number must be split evenly), the child with the lowest index receives the larger share, and the remaining budget is distributed among the others accordingly.

### 3.4 Probabilistic Modeling

We now present a formalization of our learning framework, generalizing (Bergmeister et al., 2024; Gailhard et al., 2025). Let $\{H^{(1)}, \ldots, H^{(N)}\}$ denote a set of *i.i.d.* hypergraph instances. Our objective is to approximate the unknown generative process by learning a distribution $p(H)$. We model the marginal likelihood of each hypergraph $H$ as a sum over the likelihoods of its bipartite representation's expansion sequences $p(H) = p(B) = \sum_{\varpi \in \Pi(B)} p(\varpi)$, where $\Pi(B)$ denotes the set of valid expansion sequences from a minimal bipartite graph to the full bipartite representation $B$ corresponding to $H$. Each intermediate $B^{(l-1)}$ is generated by expanding and refining its predecessor, in accordance with Definitions 2 and 3.

To simplify notations, we drop the superscript for $\mathbf{F}^{\text{refine}}$ and simply write $\mathbf{F}$. Assuming a Markovian generative structure, the likelihood of a specific expansion sequence $\varpi$ is factorized as:

$$p(\varpi) = \overbrace{p(B^{(L)})}^{1} \prod_{l=L}^{1} p(B^{(l-1)}|B^{(l)}) = \prod_{l=L}^{1} p(\mathbf{e}^{(l-1)}, \mathbf{f}^{(l-1)}, \mathbf{F}_L^{(l-1)}, \mathbf{F}_R^{(l-1)}|\tilde{B}^{(l-1)}) p(\mathbf{v}_L^{(l)}, \mathbf{v}_R^{(l)}|B^{(l)}). \tag{4}$$

To simplify the modeling process and avoid learning two separate distributions $p(\mathbf{e}^{(l)}, \mathbf{f}^{(l)}, \mathbf{F}_L^{(l)}, \mathbf{F}_R^{(l)}|\tilde{B}^{(l)})$ and $p(\mathbf{v}_L^{(l)}, \mathbf{v}_R^{(l)}|B^{(l)})$, we rearrange terms as follows:

$$p(\varpi) = p(\mathbf{e}^{(0)}, \mathbf{f}^{(0)}, \mathbf{F}_L^{(0)}, \mathbf{F}_R^{(0)}|\tilde{B}^0) p(\mathbf{v}_L^{(L)}, \mathbf{v}_R^{(L)}) \left[ \prod_{l=L-1}^{1} p(\mathbf{v}_L^{(l)}, \mathbf{v}_R^{(l)}|B^{(l)}) p(\mathbf{e}^{(l)}, \mathbf{f}^{(l)}, \mathbf{F}_L^{(l)}, \mathbf{F}_R^{(l)}|\tilde{B}^{(l)}) \right], \tag{5}$$

where $p(\mathbf{v}_L^{(L)}, \mathbf{v}_R^{(L)}) = p(\mathbf{v}_L^{(L)}, \mathbf{v}_R^{(L)}|B^{(L)})$.

We assume that the variables $\mathbf{v}_L^{(l)}$ and $\mathbf{v}_R^{(l)}$ are conditionally independent of $\tilde{B}^{(l)}$ when conditioned on $B^{(l)}$:

$$p(\mathbf{v}_L^{(l)}, \mathbf{v}_R^{(l)}|B^{(l)}, \tilde{B}^{(l)}) = p(\mathbf{v}_L^{(l)}, \mathbf{v}_R^{(l)}|B^{(l)}). \tag{6}$$

This allows us to write the combined likelihood as:

$$p(\mathbf{v}_L^{(l)}, \mathbf{v}_R^{(l)}|B^{(l)}) p(\mathbf{e}^{(l)}, \mathbf{f}^{(l)}, \mathbf{F}_L^{(l)}, \mathbf{F}_R^{(l)}|\tilde{B}^{(l)}) = p(\mathbf{v}_L^{(l)}, \mathbf{v}_R^{(l)}, \mathbf{e}^{(l)}, \mathbf{f}^{(l)}, \mathbf{F}_L^{(l)}, \mathbf{F}_R^{(l)}|\tilde{B}^{(l)}), \tag{7}$$

which corresponds to the combined expansion and refinement step that inverts coarsening at depth $l$.

During training, the model approximates the right-hand side of (7), learning to generate, for each expansion and refinement step, $\mathbf{v}_L$, $\mathbf{v}_R$, $\mathbf{e}$, $\mathbf{f}$, and $\mathbf{F}_L^{\text{refine}}$, $\mathbf{F}_R^{\text{refine}}$. Once trained, the model generates a hypergraph with $N$ nodes through successive expansion and refinement steps:

1. *Initialization.* Start from a minimal bipartite graph: $B^{(L)} = (\{1\}, \{2\}, \{(1,2)\}, (N))$, consisting of a single node on each side connected by one edge. The left-side node is assigned the full node budget, *i.e.*, $\mathbf{b} = [N]$. If node and hyperedge features need to be generated, $\mathbf{F}_L$ and $\mathbf{F}_R$ are initialized as zero matrices.

2. *Expansion and refinement.* Iteratively expand and refine the current bipartite representation to add details until the desired size is attained: $B^{(l)} \xrightarrow{\text{expand}} \tilde{B}^{(l-1)} \xrightarrow{\text{refine}} B^{(l-1)}$.

3. *Hypergraph reconstruction.* Once the final bipartite graph is generated, construct the hypergraph by collapsing each right-side node into a hyperedge connecting its adjacent left-side nodes.

## 3.5 MINIBATCH OT-COUPLING

Graphs and hypergraphs lack a natural node ordering, making it challenging to align predictions with their targets. After an expansion, the model may produce a permutation of the correct predictions; yet, without alignment, the computed loss will be large despite the prediction being correct. This mismatch introduces significant noise into the learning signal. To address this, we generalize *minibatch OT-coupling* (Tong et al., 2024; Pooladian et al., 2023) from image generation to our setting. In its original form, OT-coupling aligns samples from the prior and target distributions within a minibatch via an optimal transport plan, reindexing prior samples to minimize the matching cost. Formally, given prior samples $\mathbf{X} \in \mathbb{R}^{B \times d}$ and targets $\mathbf{Y} \in \mathbb{R}^{B \times d}$, OT-coupling finds:

$$\mathbf{P}^* = \arg\min_{\mathbf{P} \in \Pi_B} \|\mathbf{X}\mathbf{P} - \mathbf{Y}\|^2, \tag{8}$$

where $\mathbf{P}$ is a permutation matrix and $\Pi_B$ denotes the set of all $B \times B$ permutations.

In our context, for a given bipartite graph with $n$ nodes, adjacency matrix $\mathbf{A} \in \{0,1\}^{n \times n}$, expanded budgets $\mathbf{b} \in \mathbb{N}^n$ and expanded features $\mathbf{F} \in \mathbb{R}^{n \times d}$ where $d$ is the dimension of node or hyperedge features, $\mathbf{X}$ and $\mathbf{Y}$ correspond to prior samples and targets, respectively, and $\mathbf{P}$ is implemented as a node-swapping operation. To avoid bias, swaps are only allowed between *equivalent* nodes—those structurally and feature-wise indistinguishable up to noise. Swapping non-equivalent nodes would alter the learned distribution and degrade model performance. This leads to the following problem:

$$\mathbf{P}^* = \arg\min_{\mathbf{P} \in \Pi_n} \|\mathbf{P}\mathbf{X} - \mathbf{Y}\|^2 \quad \text{s.t.} \quad \mathbf{P}^\top \mathbf{A} \mathbf{P} = \mathbf{A}, \ \mathbf{P}\mathbf{b} = \mathbf{b}, \ \mathbf{P}\mathbf{F} = \mathbf{F}. \tag{9}$$

To reduce overhead, we only consider permuting children from a single cluster expansion, where equivalence holds naturally. In our implementation, with exactly two or three children per cluster, only two or six permutations are possible, making the operation lightweight and easily parallelizable with standard tensor operations.

**Proposition 2** (Informal). *Minibatch OT-coupling in our framework preserves the target distribution and produces shorter target paths.*

## 3.6 GENERALIZATION TO GRAPHS

Our framework can be directly extended to generate standard graphs by adapting the coarsen–expand procedure of Bergmeister et al. (2024). As in the hypergraph setting, each node begins with a budget of one, which is summed when nodes are merged, while features are aggregated through a weighted average with weights proportional to node budgets. During expansion, clusters propagate their budgets and features to child nodes, and the model predicts how to split the budget among them while refining their features. Graph generation, therefore, starts from a single node with a budget equal to the desired size and proceeds through successive expansion and refinement steps. To ensure stable training and alignment between predicted and target structures, minibatch OT-coupling is applied to groups of expanded nodes exactly as in the hypergraph case.

Table 1: Comparison between FAHNES and other baselines for the *SBM*, *Ego*, and *Tree* hypergraphs.

| Method | SBM Hypergraphs ($n_{avg} = 31.73, std = 0.55$) | | | Ego Hypergraphs ($n_{avg} = 109.71, std = 10.23$) | | | Tree Hypergraphs ($n_{avg} = 32, std = 0$) | | |
|---|---|---|---|---|---|---|---|---|---|
| | Valid SBM ↑ | Node Num ↓ | Spectral ↓ | Valid Ego ↑ | Node Num ↓ | Spectral ↓ | Valid Tree ↑ | Node Num ↓ | Spectral ↓ |
| HyperPA | 2.5 | 0.075 | 0.273 | 0.0 | 35.830 | 0.237 | 0.0 | 2.350 | 0.159 |
| VAE | 0.0 | 0.375 | 0.024 | 0.0 | 47.580 | 0.133 | 0.0 | 9.700 | 0.124 |
| GAN | 0.0 | 1.200 | 0.059 | 0.0 | 60.350 | 0.230 | 0.0 | 6.000 | 0.089 |
| Diffusion | 0.0 | 0.150 | 0.031 | 0.0 | 4.475 | 0.190 | 0.0 | 2.225 | 0.127 |
| HYGENE | 65.0 | 0.525 | 0.010 | 90.0 | 12.550 | **0.004** | 77.5 | **0.000** | 0.012 |
| FAHNES | **87.8±3.1** | **0.029±0.009** | **0.006±0.004** | **99.5±1.1** | 0.128±0.171 | 0.004±0.003 | **89.7±6.0** | **0.000±0.000** | **0.003±0.002** |

Table 2: Evaluation on ModelNet40.

Table 3: Evaluation for *SBM* and *Tree* graphs.

| Method | ModelNet40 Bookshelf ($n_{avg} = 119.38, std = 68.20$) | | ModelNet40 Piano ($n_{avg} = 177.29, std = 57.11$) | |
|---|---|---|---|---|
| | Node Num ↓ | Spectral ↓ | Node Num ↓ | Spectral ↓ |
| HyperPA | 8.025 | 0.048 | 0.825 | 0.067 |
| VAE | 47.450 | 0.190 | 75.350 | 0.396 |
| GAN | **0.000** | 0.476 | **0.000** | 0.697 |
| Diffusion | **0.000** | 0.079 | 0.050 | 0.113 |
| HYGENE | 69.730 | 0.068 | 42.520 | 0.117 |
| FAHNES | 0.135±0.276 | **0.024±0.015** | 0.846±1.009 | **0.040±0.026** |

| Method | SBM graphs ($n_{avg} = 105.99, std = 38.38$) | | | Tree graphs ($n_{avg} = 64.0, std = 0$) | | |
|---|---|---|---|---|---|---|
| | Valid SBM ↑ | Spectral ↓ | Ratio ↓ | Valid Tree ↑ | Spectral ↓ | Ratio ↓ |
| HSpectre | 45.0 | 0.007 | 10.2 | **100.0** | 0.012 | 4.0 |
| BwR | 7.5 | 0.017 | 38.6 | 0.0 | 0.048 | 11.4 |
| DiGress | 60.0 | **0.004** | **1.7** | 90.0 | **0.011** | **1.6** |
| DeFoG | **90.0±5.1** | 0.005±0.001 | 4.9±1.3 | 96.5±2.6 | **0.011±0.002** | 1.6±0.4 |
| FAHNES | 50.0±5.0 | **0.004±0.002** | 4.8±0.7 | **100.0±0.0** | 0.012±0.004 | 1.8±0.8 |

## 4 EXPERIMENTS AND RESULTS

In this section, we detail our experimental setup, covering datasets, evaluation metrics, implementation choices, baselines, results, and ablation studies. Full experimental details are provided in Appendix D. Visualizations of the generated samples are provided in Appendix I.

### 4.1 DATASETS AND EXPERIMENTAL SETUP

**Datasets**. We evaluate our method on five featureless hypergraph datasets: Stochastic Block Model (*SBM*) (Kim et al., 2018), *Ego* (Comrie & Kleinberg, 2021), *Tree* (Nieminen & Peltola, 1999), ModelNet40 *bookshelf*, and ModelNet40 *piano* (Wu et al., 2015). We also evaluate FAHNES on two featured hypergraphs datasets, comprising two sets of 3D meshes: Manifold40 *bench* and Manifold40 *airplane* (Hu et al., 2022). Additionally, we adapt our method to graphs using the methodology outlined in Section 3.6 and evaluate it on the five unfeatured graph datasets used in (Bergmeister et al., 2024): *SBM, Tree, Planar, Protein* and *Point cloud*. Finally, we evaluate FAHNES on two graph point cloud datasets sampled on the meshes of the two previous mesh datasets. For 3D mesh and point cloud datasets, node features are 3D positions, and edges or hyperedges do not have features.

**Metrics**. For featureless hypergraphs, we follow the evaluation criteria in (Gailhard et al., 2025). These include: *i*) structural comparison metrics such as *Node Num* (difference in node counts); *ii*) topological analysis with *Spectral* (maximum mean discrepancy between the spectral distributions). In scenarios where datasets enforce structural constraints, we report *Valid*—the percentage of generated samples satisfying those constraints. For all metrics except *Valid*, lower values indicate improved performance, while higher values are preferable for *Valid*. In the case of 3D meshes, we use *ChamDist* (the nearest training sample Chamfer distance), which computes the Chamfer distance between point clouds sampled from a generated sample and all training samples and outputs the minimum distance. For graph datasets, we report the maximum mean discrepancy (MMD) between the spectral distributions of the generated samples and training set, and the ratio of various metrics computed for the generated samples and test set (see Appendix D for details). For the the point cloud datasets, we also report the nearest training sample Chamfer distance. Detailed numerical results with additional metrics can be found in Appendix H.

**Implementation details**. Our implementation uses a custom flow-matching framework (Lipman et al., 2022) together with a *local PPGN* architecture (Bergmeister et al., 2024). We also leverage our budget component using graph inpainting techniques, enabling the model to learn only the necessary parts of the distribution: *i*) budget splits are fixed to 1 for unexpanded clusters or those with a budget of 1, *ii*) equal splits are enforced for expanded clusters with a budget of 2, *iii*) clusters of size one are not allowed to expand, and *iv*) the features of non-expanded clusters are copied unchanged.

Table 4: Evaluation on the *Planar*, *Protein* and *PointCloud* (without features) graph datasets.

| Method | Planar ($n_{avg} = 64.0$, $std = 0.0$) | | | Protein ($n_{avg} = 261.54$, $std = 104.68$) | | Point cloud ($n_{avg} = 1434.31$, $std = 1285.93$) | |
|---|---|---|---|---|---|---|---|
| | Valid Planar ↑ | Spectral ↓ | Ratio ↓ | Spectral ↓ | Ratio ↓ | Spectral ↓ | Ratio ↓ |
| HSpectre | 95.0 | 0.008 | 2.1 | **0.001** | 5.9 | **0.005** | 7.0 |
| BwR | 0.0 | 0.044 | 251.9 | 0.070 | 254.4 | 0.291 | 133.2 |
| DiGress | 77.5 | 0.010 | 5.1 | 0.002 | 18.0 | OOM | OOM |
| DeFoG | **99.5**±1.0 | **0.007**±0.001 | **1.6**±0.04 | — | — | — | — |
| FAHNES | 96.7±6.2 | 0.008±0.003 | 2.2±1.6 | **0.001**±0.001 | **3.8**±1.9 | **0.005**±0.001 | **3.2**±0.5 |

Table 5: Evaluation on the graph point cloud datasets.

| Method | Airplane Point Clouds ($n_{avg} = 971.14$, $std = 63.15$) | | | Bench Point Clouds ($n_{avg} = 987.31$, $std = 49.33$) | | |
|---|---|---|---|---|---|---|
| | Chamfer Dist ↓ | Spectral ↓ | Ratio ↓ | Chamfer Dist ↓ | Spectral ↓ | Ratio ↓ |
| DiGress | OOM | OOM | OOM | OOM | OOM | OOM |
| DeFoG | OOM | OOM | OOM | OOM | OOM | OOM |
| FAHNES | **0.094**±0.006 | **0.005**±0.004 | **67.3**±44.9 | **0.130**±0.000 | **0.004**±0.000 | **84.4**±0.0 |

Extensive implementation details are provided in Appendix E. The training and sampling procedures are explained in Appendix F. We also analyze the computational complexity in Appendix G.

**Baselines**. For the featureless hypergraphs, we compare FAHNES against HYGENE (Gailhard et al., 2025), HyperPA (Do et al., 2020), a Variational Autoencoder (VAE) (Kingma & Welling, 2013), a Generative Adversarial Network (GAN) (Goodfellow et al., 2020), and a standard 2D diffusion model (Ho et al., 2020) trained on incidence matrix images, where hyperedge membership is represented by white pixels and absence by black pixels. For 3D meshes, we compare FAHNES with a sequential disjoint generation baseline, in which the hypergraph topology is generated first, followed by the feature generation. For the graph datasets, we compare FAHNES against DeFoG (Qin et al., 2024) and DiGress (Vignac et al., 2023), two state-of-the-art flat models for graph generation, BwR (Diamant et al., 2023), a recent graph bandwidth restriction method intended for scalability, and HSpectre (Bergmeister et al., 2024), a hierarchical generation method for unfeatured graphs. All baseline results are taken from (Qin et al., 2024; Bergmeister et al., 2024).

### 4.2 RESULTS AND DISCUSSION

**Comparison with the baselines**. Tables 1 and 2 show the comparisons for the featureless hypergraphs. We see that our node-budget and OT-coupling components improve the generation quality. Regarding the Manifold40 featured hypergraph dataset[3], **FAHNES obtains a Chamfer distance of 0.064 and 0.048 for *bench* and *airplane*, respectively, while the sequential baseline reaches 0.143 and 0.117, respectively**. This shows the better modeling capability of FAHNES in jointly generating topology and features in hypergraphs instead of a simple two-step approach. About the graph datasets, Tables 3 and 4 show that FAHNES obtains competitive results compared to state-of-the-art flat methods on small-graph datasets, and that our new components improve generation quality in all datasets, compared to the method introduced in (Bergmeister et al., 2024). For point clouds, Table 5 shows that our hierarchical method is the only one able to perform at this scale, which shows its superior scalability. Detailed numerical results with more metrics are shown in Appendix H.

### 4.3 ABLATION STUDIES

Tables 6 and 7 present ablation studies of FAHNES for the node-budget and OT-coupling components. We observe that using budgets instead of concatenating the target size to each node embedding, like in (Bergmeister et al., 2024; Gailhard et al., 2025), improves generation quality. OT-coupling has a more nuanced effect, clearly improving quality on some datasets, like SBM or ManifoldNet Airplane, while not changing much for others. Those two components are also essential for feature generation,

---

[3]The full set of numerical results for the Manifold40 datasets is provided in Appendix H.

Table 6: Ablation studies on the node budget (Budg.) and minibatch OT-coupling (Coup.) for *SBM*, *Ego*, and *Tree* hypergraphs.

| Budg. | Coup. | SBM | | | Tree | | | Ego | | |
|---|---|---|---|---|---|---|---|---|---|---|
| | | Valid ↑ | NodeNum ↓ | Spectral ↓ | Valid ↑ | NodeNum ↓ | Spectral ↓ | Valid ↑ | NodeNum ↓ | Spectral ↓ |
| ✓ | ✓ | **87.8**±3.1 | **0.029**±0.009 | **0.006**±0.004 | 89.7±6.0 | **0.000**±0.000 | **0.003**±0.002 | 99.5±1.1 | 0.128±0.171 | **0.004**±0.003 |
| ✗ | ✓ | 85.3±5.9 | 0.044±0.078 | **0.006**±0.004 | 90.7±6.7 | **0.000**±0.000 | 0.004±0.001 | **100.0**±0.0 | 0.118±0.158 | 0.005±0.003 |
| ✓ | ✗ | 86.7±6.7 | 0.039±0.014 | **0.006**±0.004 | 89.3±3.6 | **0.000**±0.000 | 0.005±0.011 | 99.9±0.4 | **0.073**±0.050 | 0.007±0.012 |
| ✗ | ✗ | 84.6±4.6 | 0.049±0.045 | 0.007±0.007 | **95.6**±3.7 | **0.000**±0.000 | 0.005±0.003 | 99.5±1.1 | 0.117±0.155 | **0.004**±0.003 |

Table 7: Ablation studies on the node budget (Budg.) and minibatch OT-coupling (Coup.) for ModelNet and ManifoldNet datasets.

| Budg. | Coup. | ModelNet Bookshelf | | ModelNet Piano | | ManifoldNet Airplane | | | ManifoldNet Bench | | |
|---|---|---|---|---|---|---|---|---|---|---|---|
| | | NodeNum ↓ | Spectral ↓ | Node Num ↓ | Spectral ↓ | ChamDist ↓ | Node Num ↓ | Spectral ↓ | ChamDist ↓ | NodeNum ↓ | Spectral ↓ |
| ✓ | ✓ | **0.135**±0.276 | 0.024±0.015 | **0.846**±1.009 | 0.040±0.026 | **0.048**±0.003 | **0.017**±0.070 | **0.010**±0.003 | **0.064**±0.005 | 0.060±0.149 | 0.017±0.011 |
| ✗ | ✓ | 0.940±0.917 | 0.032±0.013 | 3.622±1.822 | 0.055±0.040 | 0.079±0.019 | 0.426±0.820 | 0.014±0.007 | 0.090±0.003 | 0.240±0.265 | 0.018±0.009 |
| ✓ | ✗ | 0.265±0.496 | **0.014**±0.007 | 3.155±3.637 | **0.030**±0.048 | 0.050±0.005 | 0.052±0.065 | 0.012±0.005 | 0.085±0.056 | **0.020**±0.032 | **0.013**±0.003 |
| ✗ | ✗ | 1.325±1.631 | 0.031±0.009 | 5.490±8.847 | 0.036±0.016 | 0.100±0.023 | 0.304±0.437 | 0.019±0.009 | 0.098±0.024 | 0.267±0.319 | 0.022±0.014 |

as lacking one of them results in much worse results, as seen by the large increase in *ChamDist* when one component is ablated.

## 4.4 LIMITATIONS

While our node budget mechanism helps mitigate the issue of missing nodes, it does not fully resolve it, and our method still struggles when generating very large hypergraphs, such as ModelNet Bookshelf and Piano, or very large graphs, such as point cloud datasets. These cases highlight that scalability remains challenging when both the number of nodes and the structural complexity grow substantially. Moreover, our framework currently assumes relatively simple feature distributions (*e.g.*, 3D coordinates), and extending it to domains with richer or heterogeneous node and hyperedge attributes (such as categorical or multi-modal features) may require additional modeling components. Finally, although FAHNES improves stability through minibatch OT-coupling, training remains computationally expensive for large-scale datasets, which may limit applicability in resource-constrained settings.

## 5 CONCLUSION

We presented FAHNES, the first scalable hierarchical framework for jointly generating graph and hypergraph topology together with features at scale. By integrating coarse-to-fine structural modeling with feature-aware generation, FAHNES overcomes the limitations of flat or disjoint approaches and enables the generation of complex structures at larger scales. Key innovations include a node budget mechanism, which provides fine-grained control over local growth, and minibatch OT-coupling, which improves alignment and stability during training. Extensive experiments on synthetic, 3D mesh, and point cloud datasets show that FAHNES achieves strong performance in both topology and feature generation while scaling to graph and hypergraph sizes that are out of reach for prior models. Our framework opens new directions for generative modeling of structured data, including applications in 3D geometry and circuit modeling, as well as extensions to richer feature modalities.

## REPRODUCIBILITY STATEMENT

All proofs of propositions and theorems, and all details regarding the experiments, parameters, and datasets can be found in the appendix. The code will be made public upon acceptance of the paper.

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

This supplementary material offers additional technical details and formal proofs to complement the main paper. The document is structured as follows: Formal proofs of the main propositions are presented in Appendix A. Appendix B describes our procedure for sampling coarsening sequences. Appendix C provides illustrative examples of coarsening sequences. Appendix D outlines the experimental setup, including hyperparameter choices and numerical results. Appendix E discusses more detailed implementation details of FAHNES. Algorithms for model training and sampling featured hypergraphs are detailed in Appendix F. Appendix G analyzes the algorithmic complexity of our approach. Appendix H presents detailed numerical results for the ablation studies. Appendix I presents visual comparisons between training and generated samples.

## A PROOFS

### A.1 AVERAGING NODE FEATURES FOR CLUSTERS' FEATURES

**Proposition 1.** *Setting cluster features as the average of the nodes they contain minimizes the Mean Squared Error (MSE) between the fully expanded hypergraph and the original hypergraph.*

*Proof.* Let $H = (\mathcal{V}, \mathcal{E})$ be a hypergraph with node set $\mathcal{V}$ and hyperedge set $\mathcal{E}$, and let $H_l = (\mathcal{V}_l, \mathcal{E}_l)$ denote the lifted hypergraph obtained by expanding each cluster $\mathcal{C} \subseteq \mathcal{V}$ of size $|C|$-clique. By construction, there exists a bijection $\phi : \mathcal{V} \to \mathcal{V}_l$ mapping each original node to its corresponding lifted node. Suppose each node $v \in \mathcal{V}$ is associated with a feature vector $\mathbf{x}_v \in \mathbb{R}^d$, and each cluster $\mathcal{C} \subseteq \mathcal{V}$ is assigned a cluster feature vector $\mathbf{x}_\mathcal{C} \in \mathbb{R}^d$, which is inherited by all lifted nodes $\phi(v)$ for $v \in \mathcal{C}$. Define the mean squared error between the original node features and the cluster features in the lifted hypergraph as:

$$\text{MSE} = \sum_{v \in \mathcal{V}} \|\mathbf{x}_v - \mathbf{x}_{C(v)}\|^2, \tag{10}$$

where $C(v)$ denotes the cluster containing node $v$.

To minimize the MSE, it suffices to minimize, for each cluster $\mathcal{C}$,

$$J_\mathcal{C}(\mathbf{x}_\mathcal{C}) = \sum_{v \in \mathcal{C}} \|\mathbf{x}_v - \mathbf{x}_\mathcal{C}\|^2. \tag{11}$$

Since $J_\mathcal{C}$ is a convex quadratic function in $\mathbf{x}_\mathcal{C}$, we find its minimum by setting the gradient to zero:

$$\nabla_{\mathbf{x}_c} J_\mathcal{C}(\mathbf{x}_C) = \sum_{v \in \mathcal{C}} 2(\mathbf{x}_\mathcal{C} - \mathbf{x}_v) = 2|\mathcal{C}|\mathbf{x}_\mathcal{C} - 2\sum_{v \in \mathcal{C}} \mathbf{x}_v = \mathbf{0}. \tag{12}$$

Solving for $\mathbf{x}_\mathcal{C}$, we obtain

$$\mathbf{x}_\mathcal{C} = \frac{1}{|\mathcal{C}|} \sum_{v \in \mathcal{C}} \mathbf{x}_v, \tag{13}$$

which is the arithmetic mean of the feature vectors in the cluster $\mathcal{C}$. $\square$

### A.2 MINIBATCH OT-COUPLING

Let $B$ be a bipartite graph. Let $\mathbf{b}$ be its current budget repartition, $\mathbf{F}_L$ its node features, and $\mathbf{F}_R$ its hyperedge features. Denote $\mathbf{x} = (\mathbf{v}_L, \mathbf{v}_R, \mathbf{e}, \mathbf{f}, \mathbf{F}_L^{\text{refine}}, \mathbf{F}_R^{\text{refine}})$, *i.e.*, the predictions of the model. In the following, all distributions are conditioned on $\mathbf{b}$, $\mathbf{F}_L$ and $\mathbf{F}_R$.

**Proposition 2** (Marginal preservation). *Under the OT-coupling of Algorithm 2, the joint distribution:*

$$q(\mathbf{x}_0, \mathbf{x}_1)$$

*has marginals:*

$$q_0(\mathbf{x}_0), \quad q_1(\mathbf{x}_1).$$

**Definition 4** (Isomorphism of bipartite graphs in our setting). Let

$$B_1 = (\mathcal{V}_L^1, \mathcal{V}_R^1, \mathcal{E}^1, \mathbf{b}^1, \mathbf{F}_L^1, \mathbf{F}_R^1), \quad B_2 = (\mathcal{V}_L^2, \mathcal{V}_R^2, \mathcal{E}^2, \mathbf{b}^2, \mathbf{F}_L^2, \mathbf{F}_R^2),$$

be bipartite graphs.

We say $B_1 \cong B_2$ (are isomorphic in our setting) if there exist bijections

$$\sigma_L \colon \mathcal{V}_L^1 \to \mathcal{V}_L^2, \quad \sigma_R \colon \mathcal{V}_R^1 \to \mathcal{V}_R^2$$

such that:

1. Edge structure is preserved: $(v, w) \in \mathcal{E}^1 \iff (\sigma_L(v), \sigma_R(w)) \in \mathcal{E}^2$,

2. Budgets are preserved: $\mathbf{b}^1(v) = \mathbf{b}^2(\sigma_L(v))$ for all $v \in \mathcal{V}_L^1$, $\quad \mathbf{b}^1(w) = \mathbf{b}^2(\sigma_R(w))$ for all $w \in \mathcal{V}_R^1$,

3. Node and hyperedge features are preserved: $\mathbf{F}_L^1(v) = \mathbf{F}_L^2(\sigma_L(v))$, $\quad \mathbf{F}_R^1(w) = \mathbf{F}_R^2(\sigma_R(w))$.

*Proof.* Let $f$ be an arbitrary test function defined on bipartite graphs. Algorithm 2 swaps noise samples between nodes that are equivalent in the sense that their target graphs (conditioned on the same topology and conditioning features) remain isomorphic after swapping. That is, if $\mathbf{x}_0$ and $\mathbf{x}_0'$ differ only by such a swap, then the resulting graphs are isomorphic: $B(\mathbf{x}_0) \cong B(\mathbf{x}_0')$. Let $\sigma$ be the bijective reindexing function corresponding to this isomorphism. Since $f$ is defined on graphs and graphs are invariant under isomorphism, we have:

$$f(\mathbf{x}_0) = f(\sigma(\mathbf{x}_0)). \tag{14}$$

Therefore:

$$\mathbb{E}_{q(\mathbf{x}_0, \mathbf{x}_1)}[f(\mathbf{x}_0)] = \mathbb{E}_{q(\mathbf{x}_1)} \left[ \mathbb{E}_{q(\mathbf{x}_0|\mathbf{x}_1)}[f(\mathbf{x}_0)] \right] \tag{15}$$

$$= \mathbb{E}_{q(\mathbf{x}_1)} \left[ \mathbb{E}_{q(\mathbf{x}_0)}[f(\sigma(\mathbf{x}_0))] \right] \tag{16}$$

$$= \mathbb{E}_{q(\mathbf{x}_1)} \left[ \mathbb{E}_{q(\mathbf{x}_0)}[f(\mathbf{x}_0)] \right] \tag{17}$$

$$= \mathbb{E}_{q(\mathbf{x}_0)}[f(\mathbf{x}_0)]. \tag{18}$$

where 16 comes from the transfer formula.

Thus, the marginal distribution over $\mathbf{x}_0$ remains unchanged. The same argument applies symmetrically for $\mathbf{x}_1$ by using $\sigma^{-1}$, concluding the proof. $\qquad\square$

**Proposition 3** (Shorter paths). *Under the OT-coupling of Algorithm 2, the loss-induced paths for a given minibatch are, on average, shorter than without OT-coupling. Formally, let $\pi$ be the permutation returned by Algorithm 2 for starting points $\mathbf{x}^0$ and target points $\mathbf{x}^1$. Then*

$$\sum_i \|\mathbf{x}_i^0 - \mathbf{x}_i^1\|^2 \geq \sum_i \|\mathbf{x}_{\pi(i)}^0 - \mathbf{x}_i^1\|^2. \tag{19}$$

*Proof.* We can rewrite the left-hand side of (19) as a sum over clusters:

$$\sum_i \|\mathbf{x}_i^0 - \mathbf{x}_i^1\|^2 = \sum_{\text{clusters } \mathcal{C}} \sum_{i \in \mathcal{C}} \|\mathbf{x}_i^0 - \mathbf{x}_i^1\|^2. \tag{20}$$

By construction, Algorithm 2 minimizes $\sum_{i \in \mathcal{C}} \|\mathbf{x}_i^0 - \mathbf{x}_i^1\|^2$ within each cluster $\mathcal{C}$.

Therefore,

$$\sum_{\text{clusters } \mathcal{C}} \sum_{i \in \mathcal{C}} \|\mathbf{x}_i^0 - \mathbf{x}_i^1\|^2 \geq \sum_{\text{clusters } \mathcal{C}} \sum_{i \in \mathcal{C}} \|\mathbf{x}_{\pi(i)}^0 - \mathbf{x}_i^1\|^2 \tag{21}$$

$$= \sum_i \|\mathbf{x}_{\pi(i)}^0 - \mathbf{x}_i^1\|^2, \tag{22}$$

which proves the claim. $\qquad\square$

## B COARSENING SEQUENCE SAMPLING

This section outlines our methodology for sampling a coarsening sequence $\pi \in \Pi_F(H)$ for a given hypergraph $H$. The full procedure is detailed in Algorithm 1. At each coarsening step $l$, let $H^{(l)}$ denote the current hypergraph, $B^{(l)}$ its bipartite representation, and $C^{(l)}$ its weighted clique expansion. We begin by sampling a target reduction fraction red_frac $\sim \mathcal{U}([\rho_{\min}, \rho_{\max}])$. We then evaluate all possible contraction sets $F(C^{(l-1)})$ using a cost function $c$, where lower cost indicates higher preference. We employ a greedy randomized strategy that processes contraction sets in order of increasing cost. For each set:

- The set is stochastically rejected with probability $1 - \lambda$.

- If not rejected:
  - **Overlap check:** If the contraction set overlaps with any previously accepted contraction, it is discarded.
  - **Coarsening attempt:** Otherwise, we compute tentative coarsened representations $C_{\text{temp}}$ and $B_{\text{temp}}$.
  - **Cluster constraint check:** If all right-side clusters in $B_{\text{temp}}$ contain at most three nodes, the contraction is accepted.
  - **Update step:** When a contraction is accepted, we:
    * Sum the budgets of the nodes in the contraction set to define the new cluster budget.
    * Compute the new cluster's node features as a weighted average of the original features, using node budgets as weights.

The loop terminates once the number of remaining nodes satisfies the stopping condition:

$$|\mathcal{V}_L^{(l-1)}| - |\bar{\mathcal{V}}_L^{(l)}| > \text{red\_frac} \cdot |\mathcal{V}_L^{(l-1)}|,$$

*i.e.*, when the number of nodes on the left side (corresponding to the original hypergraph nodes) has been reduced by the sampled fraction. This framework is flexible, allowing a variety of cost functions $c$, contraction families $F$, reduction fraction ranges $[\rho_{\min}, \rho_{\max}]$, and randomization parameters $\lambda$.

**Practical considerations.** To avoid oversampling overly small graphs during training, we follow the heuristic of Bergmeister et al. (2024): when the current graph has fewer than 16 nodes, we fix the reduction fraction to $\rho = \rho_{\max}$. Due to the constraint that no right-side cluster in $B^{(l)}$ may contain more than three nodes, achieving the target reduction fraction is not always feasible. However, we observe empirically that this rarely poses a problem when $\rho_{\max}$ is reasonably small.

During training, we sample a coarsening sequence for each dataset graph, but only retain a randomly selected intermediate graph from the sequence. Thus, our practical implementation of Algorithm 1 is designed to return a single coarsened graph with associated features and budgets, rather than the full sequence $\pi$.

To improve efficiency, we incorporate the caching mechanism introduced in Bergmeister et al. (2024). Once a coarsening sequence is generated, its levels are cached. During training, a random level is selected, returned, and then removed from the cache. A new sequence is generated only when the cache for a particular graph is depleted, avoiding unnecessary recomputation.

**Hyperparameters**. In all experiments described in Section 4, we use the following settings:

- Contraction family: The set of all edges in the clique representation, *i.e.*, $F(C) = \mathcal{E}$, for a weighted clique expansion $C = (\mathcal{V}, \mathcal{E})$.
- Cost function: Local Variation Cost Loukas (2019) with a preserving eigenspace size of $k = 8$.
- Reduction fraction range: $[\rho_{\min}, \rho_{\max}] = [0.1, 0.3]$.

---

**Algorithm 1 Hypergraph coarsening sequence sampling**: Randomized iterative coarsening of a hypergraph. At each step, contraction sets are selected based on cost, while ensuring right-side clusters never merge more than three at a time. Accepted contractions update the hypergraph structure, node budgets, and features.

---

**Parameters:** contraction family $F$, cost function $c$, reduction fraction range $[\rho_{\min}, \rho_{\max}]$, randomization parameter $\lambda$
**Input:** hypergraph $H$ with $n$ nodes and $m$ hyperedges; node features $\mathbf{F}_L$; hyperedge features $\mathbf{F}_R$
**Output:** coarsening sequence $\pi = (H^{(0)}, \ldots, H^{(L)}) \in \Pi_F(H)$

1: **function** HYPERGRAPHCOARSENINGSEQ($H$)
2:    $H^{(0)} \leftarrow H$
3:    $B^{(0)} \leftarrow$ BipartiteRepresentation($H^{(0)}$)
4:    $C^{(0)} \leftarrow$ WeightedCliqueExpansion($H^{(0)}$)
5:    $\mathbf{b}_L^{(0)} \leftarrow (1, \ldots, 1) \in \mathbb{R}^n$                                        ▷ Initial node budgets
6:    $\mathbf{b}_R^{(0)} \leftarrow (1, \ldots, 1) \in \mathbb{R}^m$                               ▷ Initial hyperedge budgets
7:    $\pi \leftarrow (B^{(0)}, \mathbf{b}_L^{(0)}, \mathbf{F}_L, \mathbf{F}_R)$
8:    $l \leftarrow 0$
9:    **while** $|\mathcal{V}_L^{(l)}| > 1$ **do**
10:      $l \leftarrow l + 1$
11:      red_frac $\sim$ Uniform($[\rho_{\min}, \rho_{\max}]$)             ▷ Sample reduction fraction
12:      $f \leftarrow c(\cdot, C^{(l-1)}, (\mathcal{P}^{(l-1)}, \ldots, \mathcal{P}^{(0)}))$           ▷ Cost function
13:      accepted_contractions $\leftarrow \emptyset$
14:      **for** $S \in$ SortedByCost($F(C^{(l-1)})$) **do**
15:        **if** Random() $> \lambda$ **then**
16:          **if** $S \cap \left(\bigcup_{P \in \text{accepted\_contractions}} P\right) = \emptyset$ **then**
17:            $C_{\text{temp}} \leftarrow$ CoarsenCliqueExpansion($C^{(l-1)}, S$)
18:            $B_{\text{temp}} \leftarrow$ CoarsenBipartite($B^{(l-1)}, S$)
19:            **if** $\forall$ right cluster $R \in B_{\text{temp}} : |R| \leq 3$ **then**
20:              accepted_contractions $\leftarrow$ accepted_contractions $\cup \{S\}$
21:              $C^{(l)} \leftarrow C_{\text{temp}}, B^{(l)} \leftarrow B_{\text{temp}}$
                                              ▷ *Update budgets and features for the new cluster*
22:              Let $S = \{v_1, \ldots, v_k\}$, and the new node be $v^*$
23:              $\mathbf{b}_L^{(l)}[v^*] \leftarrow \sum_{i=1}^{k} \mathbf{b}_L^{(l-1)}[v_i]$
24:              $\mathbf{F}_L^{(l)}[v^*] \leftarrow \frac{1}{\mathbf{b}_L^{(l)}[v^*]} \sum_{i=1}^{k} \mathbf{b}_L^{(l-1)}[v_i] \cdot \mathbf{F}_L^{(l-1)}[v_i]$
25:            **end if**
26:          **end if**
27:        **end if**
28:        **if** $|\mathcal{V}_L^{(l-1)}| - |\bar{\mathcal{V}}_L^{(l)}| >$ red_frac $\cdot |\mathcal{V}_L^{(l-1)}|$ **then**
29:          **break**
30:        **end if**
31:      **end for**
32:      $\pi \leftarrow \pi \cup \{B^{(l)}, \mathbf{b}_L^{(l)}, \mathbf{F}_L^{(l)}\}$
33:    **end while**
34:    **return** $\pi$
35: **end function**

---

## C    EXAMPLES OF COARSENING SEQUENCES

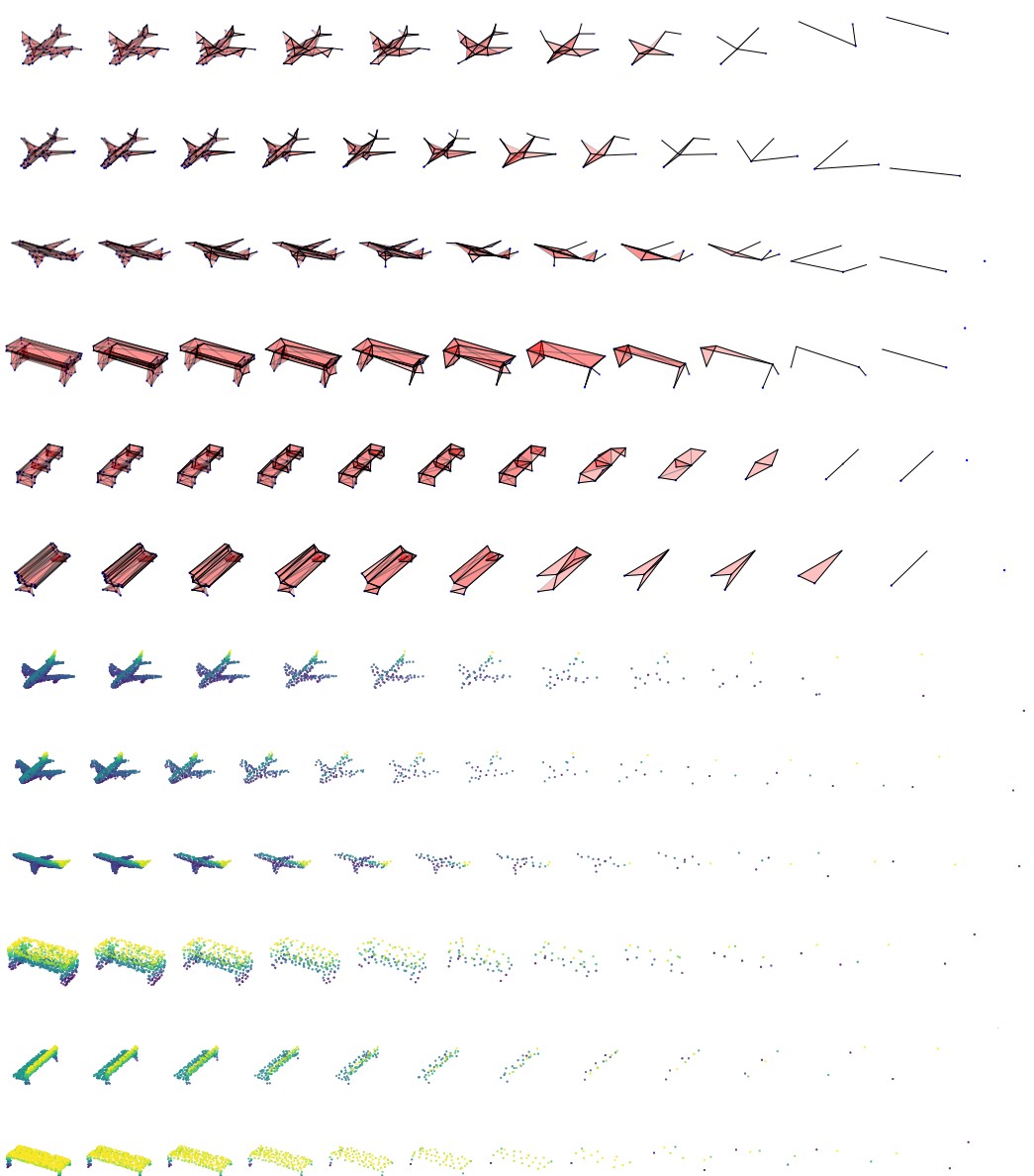

Figure 3: Examples of coarsening sequence for various meshes and point clouds. Thick lines represent 2-edges. Edges are omitted on point clouds for clarity.

## D    EXPERIMENTAL DETAILS

In this section, we detail all three types of experiments – unfeatured hypergraphs, 3D meshes and graph point clouds – individually, detailing their datasets, baselines, metrics and specific hyperparameters.

For all experiments, we use embeddings with 32 dimensions for edge selection vectors and node and hyperedge expansion numbers. When they exist, features are embedded with 128 dimensions. SignNet always has 5 layers and a hidden dimension of 128. Positional encodings always have 32 dimensions. We always use 10 layers of Local PPGN. We use the AdamW (Loshchilov & Hutter,

2017) optimizer with a learning rate of $1e-4$. All experiments are run for 1M steps on a single L40S. We use 8 CPU workers.

### D.1 UNFEATURED HYPERGRAPHS

**Datasets.** Our experiments utilize five datasets: three synthetic and two real-world, consistent with those described in Gailhard et al. (2025):

- **Stochastic Block Model (SBM) hypergraphs** Kim et al. (2018)**:** Constructed with 32 nodes split evenly into two groups. Every hyperedge connects three nodes. Hyperedges are sampled with probability 0.05 within groups and 0.001 between groups.
- **Ego hypergraphs** Comrie & Kleinberg (2021)**:** Created by generating an initial hypergraph of 150–200 nodes with 3000 randomly sampled hyperedges (up to 5 nodes each), then extracting an ego-centric subgraph by selecting a node and retaining only hyperedges that include it.
- **Tree-structured hypergraphs** Nieminen & Peltola (1999)**:** A tree with 32 nodes is generated using *networkx*, followed by grouping adjacent tree edges into hyperedges. Each hyperedge contains up to 5 nodes.
- **ModelNet40 meshes** Wu et al. (2015)**:** Hypergraphs are derived from mesh topologies of selected ModelNet40 categories. To simplify computation, meshes are downsampled to fewer than 1000 vertices by iteratively merging nearby vertices. Duplicate triangles are removed, and the resulting low-poly mesh is converted into a hypergraph. We focus on the *bookshelf* and *piano* categories.

All datasets are divided into 128 training, 32 validation, and 40 testing hypergraphs.

**Evaluation Metrics.** We evaluate generated hypergraphs using the same suite of metrics as Gailhard et al. (2025):

- **NodeNumDiff:** Average absolute difference in node count between generated and reference hypergraphs.
- **NodeDegreeDistrWasserstein:** Wasserstein distance between node degree distributions of generated and reference hypergraphs.
- **EdgeSizeDistrWasserstein:** Wasserstein distance between hyperedge size distributions.
- **Spectral:** Maximum Mean Discrepancy (MMD) between Laplacian spectra.
- **Uniqueness:** Fraction of generated hypergraphs that are non-isomorphic to one another.
- **Novelty:** Fraction of generated hypergraphs that are non-isomorphic to training samples.
- **CentralityCloseness, CentralityBetweenness, CentralityHarmonic:** Wasserstein distances computed between centrality distributions (on edges for $s = 1$). For details see Aksoy et al. (2020).
- **ValidEgo:** For the *hypergraphEgo* dataset only, proportion of generated hypergraphs that contain a central node shared by all hyperedges.
- **ValidSBM:** For the *hypergraphSBM* dataset only, proportion of generated graphs that retain the original intra- and inter-group connectivity patterns.
- **ValidTree:** For the *hypergraphTree* dataset only, proportion of generated samples that preserve tree structure.

**Baselines.** We compare our method against the following baselines:

- **HyperPA** Do et al. (2020): A heuristic approach for hypergraph generation.
- **Image-based models:** We design three baseline models—Diffusion, GAN, and VAE—that operate on incidence matrix representations of hypergraphs:
  - Each model is trained to produce binary images where white pixels signify node-hyperedge membership.
  - To normalize input sizes, incidence matrices are permuted randomly and padded with black pixels.
  - Generated images are thresholded to obtain binary incidence matrices.
- **HYGENE** Gailhard et al. (2025): A hierarchical diffusion-based generator using reduction, expansion, and refinement steps.

**Specific hyperparameters.** We use $\lambda = 0.3$, our Local PPGN layers have a dimension of 128, and the hidden dimension for our MLP is 256. We use perturbed hypergraph expansion with a radius of 2 and dropout of 0.5. The SignNet relies on the top $K = 2$ eigenvalues and eigenvectors of the graph. We use 256 sampling steps during flow-matching. Our model has 4M parameters.

## D.2 3D Meshes

**Datasets.** Datasets for meshes are taken from Manifold40 (Hu et al., 2022), which is a reworked version of ModelNet40 (Wu et al., 2015) to obtain manifold and watertight meshes. Meshes are subsequently coarsened to obtain low-poly versions of 50 vertices and 100 triangles. We use two classes:

- *Airplane* comprising 682 training samples, 21 validation samples, and 23 testing samples.
- *Bench* comprising 144 training samples, 19 validation samples, and 30 testing samples.

**Metrics.** We use the same metrics as for unfeatured hypergraphs. To this we add a metric *ChamDist* which computes the minimal Chamfer distance between a point cloud sampled from the surface of the generated mesh and equivalent point clouds sampled from all validation/test set meshes.

**Baselines.** We compare against a simple sequential baseline:

1. Our model (4M parameters) is trained for 1M steps on the topology of meshes **without** learning to generate the features.
2. A simple Local PPGN model (4M parameters) is trained for 20 epochs as a flow-matching model to learn to generate the 3D positions, with the topology fixed.
3. We use the best checkpoint of the first model to generate the topology, then apply the second model on this topology to generate the 3D positions.

**Specific hyperparameters.** We use $\lambda = 0.1$, our Local PPGN layers have a dimension of 200, and the hidden dimension for our MLP is 300. We use perturbed hypergraph expansion with a radius of 2 and dropout of 0.5. The SignNet does not rely on the eigenvalues and eigenvectors of the graph. Instead, it takes as input a random feature tensor for each node, which is replicated across all expanded nodes belonging to the same cluster. We use 25 sampling steps during flow-matching. Our model has 6M parameters.

## D.3 Unfeatured Graph Datasets

**Datasets.** We reuse the same datasets used in (Bergmeister et al., 2024):

- **Planar graphs:** This dataset consists of 200 planar graphs, each containing 64 nodes. Graphs are obtained by applying Delaunay triangulation to points sampled uniformly at random within the unit square.
- **SBM graphs:** Comprising 200 samples, these graphs are generated from a stochastic block model with 2–5 communities. Each community contains 20–40 nodes. Edges are sampled with probability 0.3 between nodes in the same community and 0.05 otherwise.
- **Tree graphs:** We generate 200 random trees with 64 nodes using *networkx*.
- **Protein graphs:** Dobson & Doig (2003) provide a dataset where proteins are represented as graphs where nodes correspond to amino acids. An edge is introduced between two nodes if the Euclidean distance between the corresponding amino acids is less than $6\,\text{Å}$. Graph sizes vary considerably.
- **Point cloud graphs:** This dataset, introduced by Liao et al. (2019), contains 41 point clouds representing household objects (Neumann et al., 2013). Each point cloud is converted into a graph, after which only the largest connected component is retained due to frequent disconnectedness in the raw graphs.

20% of samples are reserved for testing, while the remaining graphs are partitioned into 80% training and 20% validation.

**Evaluation Metrics.** We evaluate generated graphs using the same suite of metrics used in (Bergmeister et al., 2024). We compute the Maximum Mean Discrepancy (MMD) between generated and test

graphs with respect to the following graph properties *Degree distribution*, *Clustering coefficient*, *Orbit counts*, *Spectrum*, *Wavelet coefficients*. To contextualize performance, we also compute these metrics between the training and test sets, and we report the mean ratio across all applicable metrics. Certain statistics are omitted for datasets where they are ill-defined or degenerate: for the *point cloud* dataset, the degree MMD is always zero due to the $k$-nearest-neighbor construction, and for the *tree* dataset, both clustering coefficients and orbit counts are identically zero. Finally, we include dataset-specific validity metrics:

- **ValidPlanar:** Proportion of generated graphs that remain planar.
- **ValidSBM:** Proportion consistent with the original SBM parameters.
- **ValidTree:** Proportion that preserve tree structure (*i.e.*, cycle-free).

**Baselines.** We compare against two state-of-the-art flat generative models, one graph bandwidth reduction method intended for scalability, and one hierarchical generative model: Defog Qin et al. (2024), DiGress Vignac et al. (2023), BwR Diamant et al. (2023), and HSpectre Bergmeister et al. (2024).

**Specific hyperparameters.** We use $\lambda = 0.3$, our Local PPGN layers have a dimension of 128, and the hidden dimension for our MLP is 256. We use perturbed hypergraph expansion with a radius of 2 and dropout of 0.5. The SignNet relies on the top $K = 2$ eigenvalues and eigenvectors of the graph for the *Planar*, *Point cloud* and *Tree* datasets, and none for the *SBM* and *Protein* datasets. For the *Point cloud* dataset, we use dropout with probability 0.1 during the computation of node embeddings and between each layer of our GNN. We use 256 sampling steps during flow-matching. Our model has 4M parameters.

### D.4 GRAPH POINT CLOUDS

**Datasets.** We reuse the same datasets as for 3D meshes. Point clouds comprising 1024 nodes are sampled on the surface of each mesh, then each node is connected to its 3 nearest neighbors. Only the largest connected component is kept.

**Metrics. Evaluation Metrics.** We adopt the same evaluation protocol as Martinkus et al. (2022) to assess the quality of generated graphs, reporting the following metrics:

- **NumDiff:** MMD between node counts distributions of generated and reference graphs.
- **Deg:** MMD between degree distributions of generated and reference graphs.
- **Clustering:** MMD between clustering coefficient distributions.
- **Orbit:** MMD between graphlet orbit count distributions.
- **Spectral:** MMD between Laplacian spectra.
- **Wavelet:** MMD between wavelet coefficient distributions.
- **Ratio:** Average ratio of generated-to-reference values across the above metrics, used as a global indicator of statistical similarity.

To this we add a metric *ChamDist* which computes the minimal Chamfer distance between the generated mesh and all validation/test set point clouds.

**Baselines.** We compare against two state-of-the-art flat generative models: Defog Qin et al. (2024), DiGress Vignac et al. (2023). As these two methods are tailored for discrete data, we use a mixed-discrete continuous framework for continuous features.

**Specific hyperparameters.** We use $\lambda = 0.3$, our Local PPGN layers have a dimension of 128, and the hidden dimension for our MLP is 256. We use perturbed graph expansion with a radius of 2 and dropout of 0.5. The SignNet does not rely on the eigenvalues and eigenvectors of the graph. Instead, it takes as input a random feature tensor for each node, which is replicated across all expanded nodes belonging to the same cluster. We use 256 sampling steps during flow-matching. Our model has 4M parameters.

# E    IMPLEMENTATION DETAILS

## E.1    FLOW-MATCHING FRAMEWORK

We employ a flow-matching generative modeling framework Lipman et al. (2022), with endpoint parameterization following Dunn & Koes (2024), equivalent to denoising diffusion models Ho et al. (2020) using linear interpolation between the prior $p_0$ and target $p_1$ distributions. The goal is to align two distributions by transporting samples from $p_0$ to $p_1$ through a learned time-dependent vector field $\mathbf{f}(\mathbf{x}, t)$, governed by the ODE:

$$\frac{\partial \mathbf{x}}{\partial t} = \mathbf{f}(\mathbf{x}, t), \quad \mathbf{x}_0 \sim p_0. \tag{23}$$

Here, $\mathbf{f}(\mathbf{x}, t)$ is trained such that integrating this ODE produces samples from $p_1$.

**Endpoint parameterization.** Instead of directly modeling the flow field, we learn the terminal point $\hat{\mathbf{x}}_1(\mathbf{x}_t, 1)$ of the trajectory. The flow can be recovered using:

$$\mathbf{f}(\mathbf{x}_t, t) = \frac{\hat{\mathbf{x}}_1(\mathbf{x}_t, 1) - \mathbf{x}_t}{1 - t}. \tag{24}$$

**Training objective**. In classical flow-matching, the model is trained to minimize the expected squared error between the predicted and true endpoints:

$$\mathcal{L} = \mathbb{E}_{t, \mathbf{x}_0 \sim p_0, \mathbf{x}_1 \sim p_1} \left[ \|\hat{\mathbf{x}}_1(\mathbf{x}_t, 1) - \mathbf{x}_1\|^2 \right], \tag{25}$$

where $\mathbf{x}_t = (1 - t)\mathbf{x}_0 + t\mathbf{x}_1$ is a linear interpolation between samples from the prior and target distributions. For the hypergraphs, 3D mesh and point cloud datasets, during training $t$ is sampled using a log-normal distribution, which has been shown to improve performance Esser et al. (2024). For regular unfeatured graph datasets, we use a standard uniform sampling.

In our setting, each loss term is masked so that it is evaluated only on the relevant nodes or hyperedges. Specifically, we retain: *(i)* expansion losses only for nodes with budget greater than one ; *(ii)* budget losses only for expanded nodes whose parent has a budget greater than two ; and *(iii)* feature losses only for expanded nodes. For a given sample, let

$$\mathcal{L}_{\text{node-expansion}} = \|\hat{\mathbf{v}}_L - \mathbf{v}_L\|^2 \, \mathbf{1}[\mathbf{b} > 1],$$

$$\mathcal{L}_{\text{budget}} = \left\|\hat{\mathbf{f}} - \mathbf{f}\right\|^2 \mathbf{1}[\mathbf{b} > 2 \, \wedge \, \text{child of expanded cluster}],$$

$$\mathcal{L}_{\text{node-feature}} = \left\|\hat{\mathbf{F}}_L - \mathbf{F}_L^{\text{refine}}\right\|^2 \mathbf{1}[\text{child of expanded cluster}],$$

$$\mathcal{L}_{\text{hyperedge-expansion}} = \|\hat{\mathbf{v}}_R - \mathbf{v}_R\|^2,$$

$$\mathcal{L}_{\text{hyperedge-feature}} = \left\|\hat{\mathbf{F}}_R - \mathbf{F}_R^{\text{refine}}\right\|^2 \mathbf{1}[\text{child of expanded cluster}],$$

$$\mathcal{L}_{\text{incidence}} = \|\hat{\mathbf{e}} - \mathbf{e}\|^2,$$

where $\mathbf{1}[\cdot]$ denotes the indicator function. The total loss is the sum of all components, averaged across samples:

$$\mathcal{L} = \mathbb{E}[\mathcal{L}_{\text{node-expansion}} + \mathcal{L}_{\text{budget}} + \mathcal{L}_{\text{node-feature}} + \mathcal{L}_{\text{hyperedge-expansion}} + \mathcal{L}_{\text{hyperedge-feature}} + \mathcal{L}_{\text{incidence}}].$$

**Graph inpainting**. During generation, we use inpainting to enforce constraints: *i)* budget splits are fixed to 1 for unexpanded clusters or those with a budget of 1, *ii)* equal splits are enforced for expanded clusters with a budget of 2, *iii)* clusters of size one are not allowed to expand; and *iv)* features of non-expanded clusters are copied unchanged.

**Prior distributions**. We use different prior distributions depending on the task:

- **Node and edge predictions:** Prior samples $p_0$ are drawn from a Gaussian distribution. Targets are either $-1$ or $1$, indicating binary decisions (*e.g.*, whether a node is expanded or an edge is retained).

- **Hyperedge expansion:** Similar to node and edge predictions, with targets $-1$, $0$, or $1$, encoding the number of expansions (none, one, or two).
- **Budget fractions:** Prior samples $p_0$ are drawn from a Dirichlet distribution with concentration parameter $\alpha = 1.5$, linearly mapped to $[-1, 1]$ via $2x - 1$. The target is the budget fraction of each child node, linearly mapped to $[-1, 1]$ via $2x - 1$. If a cluster is not expanded, the corresponding budget becomes 1. Parent cluster budgets are encoded using sinusoidal positional encodings Vaswani et al. (2017) (dimension 32, base frequency $10^{-4}$).
- **Feature generation:** Following Ren et al. (2024), we draw prior features from a Gaussian and predict true node features, conditioned on the parent node's feature using a FiLM layer Perez et al. (2018).

**Simplex projection via Von Neumann method**. When modeling budget fractions, predictions must lie on the probability simplex. To ensure this, during generation, we project the model's outputs using the Von Neumann projection Duchi et al. (2008), which finds the closest point (in Euclidean distance) on the simplex:

$$\Delta^K = \left\{ \mathbf{x} \in \mathbb{R}^K \mid x_i \geq 0, \sum_{i=1}^{K} x_i = 1 \right\}. \tag{26}$$

*i)* Sort $\mathbf{z} \in \mathbb{R}^K$ into a descending vector $\mathbf{u}$, such that $u_1 \geq u_2 \geq \cdots \geq u_K$.

*ii)* Find the smallest index $\rho \in \{1, \ldots, K\}$ such that:

$$u_\rho - \frac{1}{\rho} \left( \sum_{j=1}^{\rho} u_j - 1 \right) > 0. \tag{27}$$

*iii)* Compute the threshold:

$$\tau = \frac{1}{\rho} \left( \sum_{j=1}^{\rho} u_j - 1 \right). \tag{28}$$

*iv)* The projection is then:

$$\mathbf{x}^* = \max(\mathbf{z} - \tau, 0). \tag{29}$$

### E.2    MODEL ARCHITECTURE

Our method represents the expansion numbers for left and right nodes, along with edge presence, as attributes of the bipartite graph. To model the distribution $p(\mathbf{v}_L^{(l)}, \mathbf{v}_R^{(l)}, \mathbf{e}^{(l)}, \mathbf{f}, \mathbf{F}_L^{\text{refine}} \mid \tilde{B}^{(l)})$, we adopt an endpoint-parameterized flow-matching framework Lipman et al. (2022). Within this framework, the attributes—namely, the expansion vectors and edge indicators—are corrupted with noise, and a denoising network is trained to reconstruct the original values.

The denoising network is structured as follows:

1. **Positional encoding:** Node positions within the graph are encoded using SignNet Lim et al. (2022). These encodings are replicated according to the respective expansion numbers.
2. **Attribute embedding:** Five separate linear layers are used to embed the bipartite graph attributes: left node features, right node features, edge features, node-specific features, and hyperedge-specific features. FiLM conditioning Perez et al. (2018) is applied to incorporate contextual information into node and hyperedge features. Node budgets are embedded using sinusoidal positional encodings Vaswani et al. (2017).
3. **Feature concatenation:**
    - For each left and right node, embeddings are concatenated with positional encodings and the desired reduction fraction. Left nodes also receive the node budget embedding.
    - If node features are present, they are appended to the left nodes. Likewise, hyperedge features are appended to the right nodes when available.
    - For edges, embeddings include edge features, concatenated positional encodings of the incident nodes, and the reduction fraction.

4. **Graph processing:** The attribute-enriched bipartite graph is processed through a stack of sparse PPGN layers, following the architecture from Bergmeister et al. (2024).

5. **Output prediction:** The final graph representations are passed through three linear projection heads to generate outputs.

   - Left node head: Predicts expansion values, budget splits, and refined node features.
   - Right node head: Predicts hyperedge expansions and refined hyperedge features.
   - Edge head: Predicts edge existence.

### E.3 ADDITIONAL DETAILS

**Perturbed expansion.** Building on Bergmeister et al. (2024); Gailhard et al. (2025), we augment Definitions 2 and 3—which are sufficient for reversing coarsening steps—with additional randomness to enhance generative quality. This modification is especially beneficial in low-data regimes where overfitting is a concern. Specifically, we introduce a probabilistic mechanism that supplements the set of edges $\tilde{\mathcal{E}}$ by randomly adding edges between node pairs on opposite sides of the bipartite graph that are within a fixed distance in $B$. The following definition extends the expansion process (Definition 2) to include this stochastic component.

**Definition 5** (Perturbed hypergraph expansion). Let $B = (\mathcal{V}_L, \mathcal{V}_R, \mathcal{E})$ be a bipartite graph, and let $\mathbf{v}_L \in \mathbb{N}^{|\mathcal{V}_L|}$ and $\mathbf{v}_R \in \mathbb{N}^{|\mathcal{V}_R|}$ denote the left and right cluster size vectors. For a given radius $r \in \mathbb{N}$ and probability $0 \leq p \leq 1$, we construct $\tilde{B}$ as in Definition 2. Additionally, for each pair of distinct nodes $\mathbf{v}_L(p) \in \tilde{\mathcal{V}}_L$ and $\mathbf{v}_R(q) \in \tilde{\mathcal{V}}_R$ that are within a distance of at most $2r + 1$ in $B$, we independently add an edge $e_{\{p_i, q_j\}}$ to $\tilde{\mathcal{E}}$ with probability $p$.

**Spectral conditioning.** In line with Martinkus et al. (2022); Bergmeister et al. (2024), we incorporate spectral information—specifically, the principal eigenvalues and eigenvectors of the *normalized* Laplacian—as a form of conditioning during the generative process. This technique has been shown to improve the quality of generated graphs. To generate $B^{(l)}$ from its coarser form $B^{(l+1)}$, we leverage the approximate spectral invariance under coarsening. We compute the $k$ smallest non-zero eigenvalues and their corresponding eigenvectors from the normalized Laplacian matrix $\mathcal{L}^{(l+1)}$ of $B^{(l+1)}$. These eigenvectors are processed using SignNet Lim et al. (2022) to produce node embeddings for $B^{(l+1)}$. These embeddings are then propagated to the expanded nodes of $B^{(l)}$, helping to preserve structural coherence and facilitate cluster identification. The hyperparameter $k$ controls the number of spectral components used.

**Minibatch OT-coupling.** Minibatch OT-coupling Tong et al. (2024); Pooladian et al. (2023) accelerates training by jointly sampling priors and targets and reindexing priors to minimize input–output distance, leading to smoother flows and fewer inference steps. As detailed in Section 3.5, we adapt this idea by treating each side of the bipartite graph as a minibatch and applying OT-coupling within them. To preserve the distribution, we restrict permutations to groups of *equivalent nodes* (identical topology and features, differing only in noise). For efficiency, we assume equivalence only among nodes from the same cluster expansion. Since each cluster produces two or three children, coupling reduces to evaluating two or six possible permutations per cluster, which can be efficiently parallelized with tensors (Algorithm 2).

---

**Algorithm 2** Minibatch OT-coupling for coarsening/expansion strategy

---

**Require:** Expanded bipartite representation $B$ with clusters $\{V^{(p)}\}$, each $V^{(p)}$ containing 1 or 2 nodes; target samples $\{x_i\}$
**Ensure:** Reindexed noise samples $\{\tilde{z}_i\}$
1: Sample noise $\{z_i\}$ for each node in $B$
2: **for all** clusters $V^{(p)} \in B$ **do**
3:     **if** $|V^{(p)}| = 1$ **then**
4:         No reassignment needed for singleton cluster
5:     **else if** $|V^{(p)}| = 2$ **then**
6:         Let $i, j$ be the indices of the two nodes in $V^{(p)}$
7:         Let $x_i, x_j$ be their corresponding targets
8:         Compute normal order cost:

$$C_{\text{normal}} \leftarrow \|z_i - x_i\|^2 + \|z_j - x_j\|^2$$

9:         Compute swapped order cost:

$$C_{\text{swap}} \leftarrow \|z_j - x_i\|^2 + \|z_i - x_j\|^2$$

10:         **if** $C_{\text{swap}} < C_{\text{normal}}$ **then**
11:             Swap $z_i \leftrightarrow z_j$
12:         **end if**
13:     **end if**
14: **end for**
15: **return** $\{\tilde{z}_i\}$ (reassigned noise samples)

---

# F  TRAINING AND SAMPLING PROCEDURES

In this section, we present the complete training and inference procedures, detailed in Algorithms 4 and 5. Both pipelines rely on node embeddings produced by Algorithm 3.

---

**Algorithm 3 Node embedding computation:** Here we describe the way the left and right side node embeddings are computed for a given bipartite representation of a hypergraph. Embeddings are computed for the input bipartite representation and then replicated according to the cluster size vectors.

---

**Parameters:** number of spectral features $k$

**Input:** bipartite representation $B = (\mathcal{V}_L, \mathcal{V}_R, \mathcal{E})$, spectral feature model SignNet$_\theta$, cluster size vector $\mathbf{v}_L$ and $\mathbf{v}_R$

**Output:** node embeddings computed for all nodes in $\mathcal{V}_L$ and $\mathcal{V}_R$ and replicated according to $\mathbf{v}_L$ and $\mathbf{v}_R$

1: **function** EMBEDDINGS($B = (\mathcal{V}_L, \mathcal{V}_R, \mathcal{E})$, SignNet$_\theta$, $\mathbf{v}_L$, $\mathbf{v}_R$)
2:     **if** $k = 0$ **then**
3:         $\mathbf{H} = [h^{(1)}, \ldots, h^{(|\mathcal{V}|)}] \overset{i.i.d.}{\sim} \mathcal{N}(0, I)$                 ▷ Sample random embeddings
4:     **else**
5:         **if** $k < |\mathcal{V}|$ **then**
6:             $[\lambda_1, \ldots, \lambda_k], [u_1, \ldots, u_k] \leftarrow \text{EIG}(B)$       ▷ Compute $k$ spectral features
7:         **else**
8:             $[\lambda_1, \ldots, \lambda_{|\mathcal{V}_L|+|\mathcal{V}_R|-1}], [u_1, \ldots, u_{|\mathcal{V}_L|+|\mathcal{V}_R|-1}] \leftarrow \text{EIG}(B)$   ▷ Compute $|\mathcal{V}_L| + |\mathcal{V}_R| - 1$ spectral features
9:             $[\lambda_{|\mathcal{V}_L|+|\mathcal{V}_R|}, \ldots, \lambda_k], [u_{|\mathcal{V}_L|+|\mathcal{V}_R|}, \ldots, u_k] \leftarrow [0, \ldots, 0], [0, \ldots, 0]$   ▷ Pad with zeros
10:         **end if**
11:         $\mathbf{H} = [h^{(1)}, \ldots, h^{(|\mathcal{V}_L|+|\mathcal{V}_R|)}] \leftarrow \text{SignNet}_\theta([\lambda_1, \ldots, \lambda_k], [u_1, \ldots, u_k], B)$
12:     **end if**
13:     $\tilde{B} = (\mathcal{V}_L^{(1)} \cup \cdots \cup \mathcal{V}_L^{(p_l)}, \mathcal{V}_R^{(1)} \cup \cdots \cup \mathcal{V}_R^{(p_r)}, \tilde{\mathcal{E}}) \leftarrow \tilde{B}(B, \mathbf{v}_L, \mathbf{v}_R)$        ▷ Expand as per Definition 2
14:     set $\tilde{B}$ s.t. for all $p_L \in [|\mathcal{V}_L|]$ and all $p_R \in [|\mathcal{V}_R|]$: for all $\mathbf{v}_L^{(p_i)} \in \mathcal{V}_L^{(p_l)}, \tilde{\mathbf{H}}[p_i] = \mathbf{H}[p_l]$ and for all $\mathbf{v}_R^{(p_i)} \in \mathcal{V}_R^{(p_r)}, \tilde{\mathbf{H}}[p_i] = \mathbf{H}[p_r]$          ▷ Replicate embeddings
15:     **return** $\tilde{\mathbf{H}}$
16: **end function**

---

---

**Algorithm 4 End-to-end training procedure:** This describes the entire training procedure for our model.

---

**Parameters:** number of spectral features $k$ for node embeddings
**Input:** dataset $\mathcal{D} = \{H_1, \ldots, H_N\}$, denoising model $\text{GNN}_\theta$, spectral feature model $\text{SignNet}_\theta$
**Output:** trained model parameters $\theta$
1: **function** TRAIN($\mathcal{D}$, $\text{GNN}_\theta$, $\text{SignNet}_\theta$)
2:     **while** not converged **do**
3:         $H \sim \text{Uniform}(\mathcal{D})$                                           $\triangleright$ Sample graph
4:         $(B^{(0)}, \ldots, B^{(L)}) \leftarrow \text{RndRedSeq}(H)$     $\triangleright$ Sample coarsening sequence by Algorithm 1
5:         $l \sim \text{Uniform}(\{0, \ldots, L\})$                                $\triangleright$ Sample level
6:         **if** $l = 0$ **then**
7:             $\mathbf{v}_L^{(0)} \leftarrow 1, \mathbf{v}_R^{(0)} \leftarrow 1$
8:         **else**
9:             set $\mathbf{v}_L^{(l)}$ and $\mathbf{v}_R^{(l)}$ such that the node sets of $\tilde{B}(B^{(l)}, \mathbf{v}_L^{(l)}, \mathbf{v}_R^{(l)})$ equals that of $B^{(l-1)}$
10:        **end if**
11:        **if** $l = L$ **then**
12:           $B^{(l+1)} \leftarrow B^{(l)} = (\{1\}, \{2\}, \{(1,2)\}, \mathbf{b} = size(H), \mathbf{F}_L = 0, \mathbf{F}_R = 0)$
13:           $\mathbf{v}_L^{(l+1)} \leftarrow 1$
14:           $\mathbf{v}_R^{(l+1)} \leftarrow 1$
15:           $\mathbf{e}^{(l)} \leftarrow 1$
16:        **end if**
17:        set $\mathbf{e}^{(l)}, \mathbf{f}, \mathbf{F}_L^{\text{refine}}$ and $\mathbf{F}_R^{\text{refine}}$ such that $B(\tilde{B}(B^{(l+1)}, \mathbf{v}_L^{(l+1)}, \mathbf{v}_R^{(l+1)}), \mathbf{e}^{(l)}, \mathbf{f}, \mathbf{F}_L^{\text{refine}}, \mathbf{F}_R^{\text{refine}}) = B^{(L)}$
18:         $\mathbf{H}^{(l)} \leftarrow \text{Embeddings}(B^{(l+1)}, \text{SignNet}_\theta, \mathbf{v}_L^{(l+1)}, \mathbf{v}_R^{(l+1)})$    $\triangleright$ Compute node embeddings
19:         $\hat{\rho} \leftarrow 1 - (n^{(l)}/n^{(l-1)})$, with $n^{(l)}$ and $n^{(l-1)}$ being the size of the left side of $B^{(l)}$ and $B^{(l-1)}$
20:         $D_\theta \leftarrow \text{GNN}_\theta(\cdot, \cdot, \tilde{B}^{(l)}, \mathbf{H}^{(l)}, n^{(0)}, \rho)$, where $n^{(0)}$ is the size of the left side of $B^{(0)}$
21:         take gradient descent step on $\nabla_\theta \text{DiffusionLoss}(\mathbf{v}_L^{(L)}, \mathbf{v}_R^{(L)}, \mathbf{e}^{(l)}, \mathbf{f}, \mathbf{F}_L^{\text{refine}}, \mathbf{F}_R^{\text{refine}}, D_\theta)$
22:     **end while**
23:     **return** $\theta$
24: **end function**

---

**Algorithm 5 End-to-end sampling procedure with deterministic expansion size:** This describes the sampling procedure. Note that this assumes that the maximum cluster sizes are 2 and 3, which is the case when using edges of the clique representation as the contraction set family for model training.

---

**Parameters:** reduction fraction range $[\rho_{\min}, \rho_{\max}]$
**Input:** target hypergraph size $N$, denoising model $\text{GNN}_\theta$, spectral feature model $\text{SignNet}_\theta$
**Output:** sampled hypergraph $H = (\mathcal{V}, \mathcal{E})$ with $|\mathcal{V}| = N$

1: **function** SAMPLE($N$, $\text{GNN}_\theta$, $\text{SignNet}_\theta$)
2:  $\quad B = (\mathcal{V}_L, \mathcal{V}_R, \mathcal{E}, \mathbf{f}, \mathbf{F}_L^{\text{refine}}, \mathbf{F}_R^{\text{refine}}) \leftarrow (\{1\}, \{2\}, \{(1,2)\}, N, 0, 0)$  $\quad\triangleright$ Start with a minimal bipartite graph
3:  $\quad \mathbf{v}_L \leftarrow [1], \mathbf{v}_R \leftarrow [1]$  $\quad\triangleright$ Initial cluster size vectors
4:  $\quad$ **while** $|\mathcal{V}_L| < N$ **do**
5:  $\quad\quad \mathbf{H} \leftarrow \text{Embeddings}(B, \text{SignNet}_\theta, \mathbf{v}_L, \mathbf{v}_R)$  $\quad\triangleright$ Compute node embeddings
6:  $\quad\quad n \leftarrow \|\mathbf{v}_L\|_1$
7:  $\quad\quad \rho \sim \text{Uniform}([\rho_{\min}, \rho_{\max}])$  $\quad\triangleright$ random reduction fraction
8:  $\quad\quad$ set $n^+$ s.t. $n^+ = \lceil \rho(n + n^+) \rceil$  $\quad\triangleright$ number of left side nodes to add
9:  $\quad\quad n^+ \leftarrow \min(n^+, N - n)$  $\quad\triangleright$ ensure not to exceed target size
10:  $\quad\quad \hat{\rho} \leftarrow 1 - (n/(n + n^+))$  $\quad\triangleright$ actual reduction fraction
11:  $\quad\quad D_\theta \leftarrow \text{GNN}_\theta(\cdot, \cdot, \tilde{B}(B, \mathbf{v}_L, \mathbf{v}_R), \mathbf{H}, N, \hat{\rho})$
12:  $\quad\quad (\mathbf{v}_L)_0, (\mathbf{v}_R)_0, (\mathbf{e})_0, \mathbf{f}, \mathbf{F}_L^{\text{refine}}, \mathbf{F}_R^{\text{refine}} \leftarrow \text{Sample}(D_\theta)$  $\quad\triangleright$ Sample features
13:  $\quad\quad$ set $\mathbf{v}_L$ s.t. for $i \in [n]$: $\mathbf{v}_L[i] = 2$ if $|\{j \in [n] \mid (\mathbf{v}_L)_0[j] \geq (\mathbf{v}_L)_0[i]\}| \geq n^+$ and $v[i] = 1$ otherwise
14:  $\quad\quad$ set $\mathbf{v}_R$ s.t. for $i \in [|(\mathbf{v}_R)_0|]$: $\mathbf{v}_R[i] = 1$ if $(\mathbf{v}_R)_0 < 1.66$, $\mathbf{v}_R[i] = 2$ if $(\mathbf{v}_R)_0 < 2.33$ and $\mathbf{v}_R[i] = 3$ otherwise
15:  $\quad\quad$ set $\mathbf{e}$ s.t. for $i \in [|(\mathbf{e})_0|]$: $\mathbf{e}[i] = 1$ if $(\mathbf{e})_0 > 0.5$ and $\mathbf{e}[i] = 0$ otherwise
16:  $\quad\quad B = (\mathcal{V}_L, \mathcal{V}_R, \mathcal{E}) \leftarrow B(\tilde{B}, \mathbf{e}, \mathbf{f}, \mathbf{F}_L^{\text{refine}}, \mathbf{F}_R^{\text{refine}})$  $\quad\triangleright$ Refine as per Definition 3
17:  $\quad$ **end while**
18:  $\quad$ build $H$ from its bipartite representation $B$
19:  $\quad$ **return** $H$
20: **end function**

---

## G COMPLEXITY ANALYSIS

In this section, we investigate the asymptotic complexity of our proposed algorithm, which extends the methodology introduced by Bergmeister et al. (2024) and Gailhard et al. (2025). To construct a hypergraph comprising $n$ nodes, $m$ hyperedges, and $k$ incidences, the algorithm sequentially produces a series of bipartite graphs $B^{(L)} = (\{1\}, \{2\}, \{(1, 2)\}), B^{(L-1)}, \ldots, B^{(0)} = B$, where the final graph $B$ corresponds to the bipartite representation of the generated hypergraph. We use $n$, $m$, and $k$ to denote, respectively, the number of nodes, hyperedges, and incidences in the hypergraph, and as the number of left-side nodes, right-side nodes, and edges in the corresponding bipartite graph.

For each level $0 \leq l < L$ of the sequence, the number of left-side nodes in $B^{(l)}$, denoted $n_l$, satisfies $n_l \geq (1 + \epsilon)n_{l-1}$ for some $\epsilon > 0$ (*e.g.*, $\epsilon = \texttt{reduction\_frac}/(1 - \texttt{reduction\_frac})$). This implies an upper bound on the number of steps in the expansion sequence: $\lceil \log_{1+\epsilon} n \rceil \in \mathcal{O}(\log n)$. Since the expansion process only increases node counts, all $B_l$ graphs contain fewer than $n$ left-side and $m$ right-side nodes. The number of edges, however, may temporarily exceed $k$, as the intermediate bipartite graphs may include additional edges removed in later refinements. Still, because the coarsening during training consistently reduces incidences, the model is expected to learn accurate edge refinement and avoid such accumulation. Consequently, we assume $k_l \leq k$ and $m_l \leq m$ for all $0 \leq l \leq L$.

Next, we assess the computational cost of generating a single expansion step. At level $l = L$, this consists of creating a pair of connected nodes, initializing features as matrices of zeros, initializing budget as the targeted node count, and predicting the expansion vectors $\mathbf{v}_L$ and $\mathbf{v}_R$—a process with constant complexity $\mathcal{O}(1)$. For levels $0 \leq l < L$, given $B^{(l+1)}$ and expansion vectors $\mathbf{v}_L^{(l+1)}$, $\mathbf{v}_R^{(l+1)}$, the algorithm constructs the expanded bipartite graph $\tilde{B}(B^{(l+1)}, \mathbf{v}_L^{(l+1)}, \mathbf{v}_R^{(l+1)})$ in $\mathcal{O}(n + m)$ time. It then samples $\mathbf{v}_L^{(l)}, \mathbf{v}_R^{(l)}, \mathbf{e}^{(l)}, \mathbf{f}, \mathbf{F}_L^{\text{refine}}$, and $\mathbf{F}_R^{\text{refine}}$, and constructs the refined graph $B^{(l)} = B(\tilde{B}^{(l)}, \mathbf{e}^{(l)}, \mathbf{f}, \mathbf{F}_L^{\text{refine}}, \mathbf{F}_R^{\text{refine}})$. Letting $v_{\max}^L$ and $v_{\max}^R$ be the maximum cluster sizes, the incidence count in $\tilde{B}^{(l)}$ is bounded by $k_l \leq k_{l+1} v_{\max}^L v_{\max}^R$.

The sampling process queries a denoising model a constant number of times per step. The complexity is thus governed by the architecture. In our case, since bipartite graphs are triangle-free, the *Local PPGN* model Bergmeister et al. (2024) has linear complexity $\mathcal{O}(n + m + k)$. Embedding computation for $B^{(l)}$ similarly costs $\mathcal{O}(n + m + k)$. This includes calculating the top $K$ eigenvalues/eigenvectors of the Laplacian via the method from Vishnoi (2013), with complexity $\mathcal{O}\left(K(n_{l+1} + m_{l+1} + k_{l+1})\right)$, and embedding via *SignNet*, also linear in graph size due to fixed $K$.

The final transformation from the bipartite graph to a hypergraph—by collapsing right-side nodes into hyperedges—has a cost of $\mathcal{O}(m + k)$. Under these assumptions, the total complexity to generate a hypergraph $H$ with $n$ nodes, $m$ hyperedges, and $k$ incidences is $\mathcal{O}(n + m + k)$.

# H DETAILED NUMERICAL RESULTS

In this section, we present detailed numerical results for all datasets and metrics described in Appendix D. Reported values of the form $a \pm b$ indicate the mean $a$ and twice the standard deviation $b$ computed over 5 runs. The best and second-best results are highlighted in **bold** and underlined, respectively.

## H.1 COMPARISONS WITH THE BASELINES

| Method | Valid ↑ | NumDiff ↓ | Deg ↓ | EdgeSize ↓ | Spectral ↓ | Harmonic ↓ | Closeness ↓ | Betweenness ↓ |
|---|---|---|---|---|---|---|---|---|
| **SBM Hypergraphs** | | | | | | | | |
| HyperPA Do et al. (2020) | 2.5 | 0.075 | 4.062 | 0.407 | 0.273 | 77.840 | 0.074 | 0.008 |
| VAE Kingma & Welling (2013) | 0.0 | 0.375 | 1.280 | 1.059 | 0.024 | 6.543 | **0.007** | 0.006 |
| GAN Goodfellow et al. (2020) | 0.0 | 1.200 | 2.106 | 1.203 | 0.059 | 10.700 | 0.076 | 0.012 |
| Diffusion Ho et al. (2020) | 0.0 | 0.150 | 1.717 | 1.390 | 0.031 | 13.940 | 0.040 | 0.004 |
| HYGENE Gailhard et al. (2025) | 65.0 | 0.525 | **0.321** | **0.002** | 0.010 | 2.990 | 0.016 | **0.000** |
| FAHNES | **87.8±3.1** | **0.029±0.009** | 0.846±0.457 | 0.005±0.003 | **0.006±0.004** | 6.410±3.124 | 0.009±0.006 | 0.003±0.001 |
| **Ego Hypergraphs** | | | | | | | | |
| HyperPA Do et al. (2020) | 0.0 | 35.830 | 2.590 | 0.423 | 0.237 | 143.000 | 0.354 | 0.002 |
| VAE Kingma & Welling (2013) | 0.0 | 47.580 | 0.803 | 1.458 | 0.133 | 38.950 | 0.558 | 0.019 |
| GAN Goodfellow et al. (2020) | 0.0 | 60.350 | 0.917 | 1.665 | 0.230 | 41.800 | 0.612 | 0.015 |
| Diffusion Ho et al. (2020) | 0.0 | 4.475 | 3.984 | 2.985 | 0.190 | 6.911 | 0.407 | 0.009 |
| HYGENE Gailhard et al. (2025) | 90.0 | 12.550 | **0.063** | 0.220 | **0.004** | 5.790 | 0.025 | **0.000** |
| FAHNES | **99.5±1.1** | **0.128±0.171** | 0.124±0.086 | **0.155±0.067** | **0.004±0.003** | **2.703±3.468** | **0.003±0.005** | **0.000±0.000** |
| **Tree Hypergraphs** | | | | | | | | |
| HyperPA Do et al. (2020) | 0.0 | 2.350 | 0.315 | 0.284 | 0.159 | 5.941 | 0.477 | 0.168 |
| VAE Kingma & Welling (2013) | 0.0 | 9.700 | 0.072 | 0.480 | 0.124 | 3.869 | 0.280 | 0.139 |
| GAN Goodfellow et al. (2020) | 0.0 | 6.000 | 0.151 | 0.469 | 0.089 | 2.198 | 0.201 | 0.124 |
| Diffusion Ho et al. (2020) | 0.0 | 2.225 | 1.718 | 1.922 | 0.127 | 8.565 | 0.353 | 0.139 |
| HYGENE Gailhard et al. (2025) | 77.5 | **0.000** | 0.059 | 0.108 | 0.012 | 1.099 | 0.041 | 0.016 |
| FAHNES | **89.7±6.0** | **0.000±0.000** | **0.022±0.022** | **0.030±0.034** | **0.003±0.002** | **0.171±0.106** | **0.014±0.006** | **0.014±0.004** |

| Method | NumDiff ↓ | Deg ↓ | EdgeSize ↓ | Spectral ↓ | Harmonic ↓ | Closeness ↓ | Betweenness ↓ |
|---|---|---|---|---|---|---|---|
| **ModelNet Bookshelf** | | | | | | | |
| HyperPA Do et al. (2020) | 8.025 | 7.562 | 0.044 | 0.048 | 877.500 | 0.211 | 0.005 |
| VAE Kingma & Welling (2013) | 47.450 | 6.190 | 1.520 | 0.190 | 113.600 | 0.145 | 0.003 |
| GAN Goodfellow et al. (2020) | **0.000** | 397.200 | 46.300 | 0.476 | 670.100 | 0.707 | 0.007 |
| Diffusion Ho et al. (2020) | **0.000** | 20.360 | 2.346 | 0.079 | 264.100 | 0.239 | 0.006 |
| HYGENE Gailhard et al. (2025) | 69.730 | **1.050** | 0.034 | 0.068 | **27.400** | 0.204 | 0.004 |
| FAHNES | 0.135±0.276 | 2.980±2.107 | **0.020±0.025** | **0.024±0.015** | 46.614±58.803 | **0.086±0.030** | **0.001±0.001** |
| **ModelNet Piano** | | | | | | | |
| HyperPA Do et al. (2020) | 0.825 | 9.254 | **0.023** | 0.067 | **77.840** | 0.236 | 0.004 |
| VAE Kingma & Welling (2013) | 75.350 | 8.060 | 1.686 | 0.396 | 184.300 | 0.241 | 0.003 |
| GAN Goodfellow et al. (2020) | **0.000** | 409.000 | 86.380 | 0.697 | 622.200 | 0.738 | 0.005 |
| Diffusion Ho et al. (2020) | 0.050 | 20.900 | 4.192 | 0.113 | 289.300 | 0.303 | 0.004 |
| HYGENE Gailhard et al. (2025) | 42.520 | 6.290 | 0.027 | 0.117 | 155.000 | 0.285 | **0.002** |
| FAHNES | 0.846±1.009 | **3.265±1.954** | 0.042±0.056 | **0.040±0.026** | 119.158±81.023 | **0.123±0.119** | **0.002±0.002** |

Table 8: Detailed numerical results for unfeatured hypergraphs.

| Method | ChamDist | NumDiff ↓ | Deg ↓ | EdgeSize ↓ | Spectral ↓ | Harmonic ↓ | Closeness ↓ | Betweenness ↓ |
|---|---|---|---|---|---|---|---|---|
| **ManifoldNet Airplane** | | | | | | | | |
| Sequential | 0.143 | 0.367 | 0.801 | **0.004** | **0.007** | 4.964 | 0.087 | 0.006 |
| FAHNES | **0.048±0.003** | **0.017±0.070** | **0.218±0.085** | **0.004±0.007** | 0.010±0.003 | **2.428±1.455** | **0.032±0.007** | **0.003±0.001** |
| **ManifoldNet Bench** | | | | | | | | |
| Sequential | 0.117 | 0.078 | **0.332** | 0.011 | **0.015** | **3.131** | **0.035** | 0.007 |
| FAHNES | **0.064±0.005** | **0.060±0.149** | 0.349±0.454 | **0.009±0.017** | 0.017±0.011 | 5.069±8.171 | 0.046±0.040 | **0.004±0.003** |

Table 9: Detailed numerical results for 3D meshes.

| SBM Graphs | | | | | | | |
|---|---|---|---|---|---|---|---|
| **Method** | **Valid ↑** | **Wavelet ↓** | **Orbit ↓** | **Clustering ↓** | **Deg ↓** | **Spectral ↓** | **Ratio ↓** |
| DiGress Vignac et al. (2023) | 60.0 | **0.001** | 0.042 | **0.049** | 0.002 | **0.004** | **1.7** |
| DeFoG Qin et al. (2024) | **90.0**±5.1 | 0.008±0.002 | 0.056±0.074 | 0.052±0.001 | **0.001**±0.002 | 0.005±0.001 | 4.9±1.3 |
| HSpectre Bergmeister et al. (2024) | 45.0 | 0.022 | 0.067 | 0.052 | 0.012 | 0.007 | 10.2 |
| BwR (Diamant et al., 2023) | 7.5 | 0.089 | 0.114 | 0.064 | 0.048 | 0.0169 | 38.6 |
| FAHNES | 50.0±5.0 | 0.008±0.003 | **0.040**±0.000 | 0.052±0.004 | 0.007±0.000 | **0.004**±0.002 | 4.8±0.7 |
| Tree Graphs | | | | | | | |
| **Method** | **Valid ↑** | **Wavelet ↓** | **Orbit ↓** | **Clustering ↓** | **Deg ↓** | **Spectral ↓** | **Ratio ↓** |
| DiGress Vignac et al. (2023) | 90.0 | **0.004** | **0.000** | **0.000** | **0.000** | **0.011** | **1.6** |
| DeFoG Qin et al. (2024) | 96.5±2.6 | 0.005±0.000 | 0.000±0.000 | 0.000±0.000 | 0.000±0.000 | **0.011**±0.003 | **1.6**±0.4 |
| HSpectre Bergmeister et al. (2024) | **100.0** | 0.005 | 0.000 | 0.000 | 0.000 | 0.012 | 4.0 |
| BwR (Diamant et al., 2023) | 0.0 | 0.039 | 0.000 | 0.124 | 0.002 | 0.048 | 11.4 |
| FAHNES | **100.0**±0.0 | 0.005±0.001 | **0.000**±0.000 | **0.000**±0.000 | **0.000**±0.000 | 0.012±0.004 | 1.8±0.8 |
| Planar Graphs | | | | | | | |
| **Method** | **Valid ↑** | **Wavelet ↓** | **Orbit ↓** | **Clustering ↓** | **Deg ↓** | **Spectral ↓** | **Ratio ↓** |
| DiGress Vignac et al. (2023) | 77.5 | 0.003 | 0.008 | 0.078 | 0.001 | 0.010 | 5.1 |
| DeFoG Qin et al. (2024) | **99.5**±1.0 | **0.001**±0.000 | **0.001**±0.000 | 0.050±0.015 | **0.001**±0.000 | **0.007**±0.001 | **1.6**±0.4 |
| HSpectre Bergmeister et al. (2024) | 95.0 | **0.001** | 0.002 | 0.063 | **0.001** | 0.007 | 2.1 |
| BwR (Diamant et al., 2023) | 0.0 | 0.131 | 0.547 | 0.260 | 0.023 | 0.044 | 251.9 |
| FAHNES | 96.7±6.2 | **0.001**±0.001 | 0.003±0.005 | **0.042**±0.002 | **0.000**±0.000 | 0.008±0.003 | 2.2±1.6 |
| Protein Graphs | | | | | | | |
| **Method** | | **Wavelet ↓** | **Orbit ↓** | **Clustering ↓** | **Deg ↓** | **Spectral ↓** | **Ratio ↓** |
| DiGress Vignac et al. (2023) | | 0.006 | 0.129 | 0.049 | 0.004 | 0.002 | 18.0 |
| HSpectre Bergmeister et al. (2024) | | 0.003 | **0.005** | **0.031** | 0.003 | **0.001** | 5.9 |
| BwR (Diamant et al., 2023) | | 0.120 | 0.494 | 0.420 | 0.126 | 0.070 | 245.4 |
| FAHNES | | **0.001**±0.001 | 0.013±0.004 | 0.039±0.005 | **0.000**±0.001 | **0.001**±0.001 | **3.8**±1.9 |
| Point Cloud (Unfeatured) Graphs | | | | | | | |
| **Method** | | **Wavelet ↓** | **Orbit ↓** | **Clustering ↓** | **Deg ↓** | **Spectral ↓** | **Ratio ↓** |
| DiGress Vignac et al. (2023) | | OOM | OOM | OOM | OOM | OOM | OOM |
| HSpectre Bergmeister et al. (2024) | | 0.019 | 0.078 | 0.578 | 0.014 | **0.005** | 7.0 |
| BwR (Diamant et al., 2023) | | 0.592 | 1.073 | 0.469 | 0.493 | 0.291 | 133.2 |
| FAHNES | | **0.018**±0.005 | **0.009**±0.001 | **0.441**±0.073 | **0.006**±0.012 | 0.006±0.001 | **3.2**±0.5 |

Table 10: Detailed numerical results for unfeatured graph datasets.

| Airplane Point Clouds | | | | | | | | |
|---|---|---|---|---|---|---|---|---|
| **Method** | **ChamDist ↓** | **NumDiff ↓** | **Wavelet ↓** | **Orbit ↓** | **Clustering ↓** | **Deg ↓** | **Spectral ↓** | **Ratio ↓** |
| DiGress Vignac et al. (2023) | OOM | OOM | OOM | OOM | OOM | OOM | OOM | OOM |
| DeFoG Qin et al. (2024) | OOM | OOM | OOM | OOM | OOM | OOM | OOM | OOM |
| FAHNES | 0.094±0.006 | 0.000±0.000 | 0.004±0.001 | 0.074±0.104 | 0.264±0.254 | 0.002±0.002 | 0.005±0.004 | 67.3±44.9 |
| Bench Point Clouds | | | | | | | | |
| **Method** | **ChamDist ↓** | **NumDiff ↓** | **Wavelet ↓** | **Orbit ↓** | **Clustering ↓** | **Deg ↓** | **Spectral ↓** | **Ratio ↓** |
| DiGress Vignac et al. (2023) | OOM | OOM | OOM | OOM | OOM | OOM | OOM | OOM |
| DeFoG Qin et al. (2024) | OOM | OOM | OOM | OOM | OOM | OOM | OOM | OOM |
| FAHNES | 0.130±0.001 | 0.000±0.000 | 0.003±0.000 | 0.013±0.004 | 0.229±0.071 | 0.000±0.000 | 0.004±0.000 | 73.3±22.2 |

Table 11: Detailed numerical results for graph point cloud datasets.

## H.2 ABLATION STUDIES

| Node Budget | Minibatch OT | Valid ↑ | NumDiff ↓ | Deg ↓ | EdgeSize ↓ | Spectral ↓ | Harmonic ↓ | Closeness ↓ | Betweenness ↓ |
|---|---|---|---|---|---|---|---|---|---|
| | | | | **SBM Hypergraphs** | | | | | |
| ✓ | ✓ | $\mathbf{87.8}_{\pm3.1}$ | $\mathbf{0.029}_{\pm0.009}$ | $\mathbf{0.846}_{\pm0.457}$ | $\mathbf{0.005}_{\pm0.003}$ | $\mathbf{0.006}_{\pm0.004}$ | $\underline{6.410}_{\pm3.124}$ | $\mathbf{0.009}_{\pm0.006}$ | $\mathbf{0.003}_{\pm0.001}$ |
| ✗ | ✓ | $85.3_{\pm5.9}$ | $0.044_{\pm0.078}$ | $\underline{0.856}_{\pm0.636}$ | $0.023_{\pm0.022}$ | $\mathbf{0.006}_{\pm0.004}$ | $\mathbf{6.210}_{\pm4.649}$ | $\mathbf{0.009}_{\pm0.013}$ | $\mathbf{0.003}_{\pm0.003}$ |
| ✓ | ✗ | $\underline{86.7}_{\pm6.7}$ | $0.039_{\pm0.014}$ | $0.910_{\pm0.363}$ | $0.006_{\pm0.004}$ | $\mathbf{0.006}_{\pm0.004}$ | $6.815_{\pm1.913}$ | $0.010_{\pm0.005}$ | $0.004_{\pm0.001}$ |
| ✗ | ✗ | $84.6_{\pm4.6}$ | $0.049_{\pm0.045}$ | $0.916_{\pm0.749}$ | $0.047_{\pm0.070}$ | $0.007_{\pm0.007}$ | $6.665_{\pm5.199}$ | $0.012_{\pm0.014}$ | $0.004_{\pm0.004}$ |
| | | | | **Ego Hypergraphs** | | | | | |
| ✓ | ✓ | $99.5_{\pm1.1}$ | $0.128_{\pm0.171}$ | $\underline{0.124}_{\pm0.086}$ | $0.155_{\pm0.067}$ | $\mathbf{0.004}_{\pm0.003}$ | $2.703_{\pm3.468}$ | $0.003_{\pm0.005}$ | $\mathbf{0.000}_{\pm0.000}$ |
| ✗ | ✓ | $\mathbf{100.0}_{\pm0.0}$ | $0.118_{\pm0.158}$ | $0.140_{\pm0.104}$ | $\mathbf{0.133}_{\pm0.106}$ | $0.005_{\pm0.003}$ | $3.426_{\pm2.629}$ | $\mathbf{0.000}_{\pm0.000}$ | $\mathbf{0.000}_{\pm0.000}$ |
| ✓ | ✗ | $\underline{99.9}_{\pm0.4}$ | $\mathbf{0.073}_{\pm0.050}$ | $0.237_{\pm0.465}$ | $0.193_{\pm0.175}$ | $0.007_{\pm0.012}$ | $4.521_{\pm5.629}$ | $0.000_{\pm0.002}$ | $\mathbf{0.000}_{\pm0.000}$ |
| ✗ | ✗ | $99.5_{\pm1.1}$ | $\underline{0.117}_{\pm0.155}$ | $0.110_{\pm0.111}$ | $0.189_{\pm0.084}$ | $\mathbf{0.004}_{\pm0.003}$ | $\underline{2.765}_{\pm1.684}$ | $0.001_{\pm0.004}$ | $\mathbf{0.000}_{\pm0.000}$ |
| | | | | **Tree Hypergraphs** | | | | | |
| ✓ | ✓ | $89.7_{\pm6.0}$ | $\mathbf{0.000}_{\pm0.000}$ | $\underline{0.022}_{\pm0.022}$ | $\mathbf{0.030}_{\pm0.034}$ | $\mathbf{0.003}_{\pm0.002}$ | $\underline{0.171}_{\pm0.106}$ | $\underline{0.014}_{\pm0.006}$ | $0.014_{\pm0.004}$ |
| ✗ | ✓ | $\underline{90.7}_{\pm6.7}$ | $\mathbf{0.000}_{\pm0.000}$ | $0.024_{\pm0.019}$ | $0.067_{\pm0.016}$ | $\underline{0.004}_{\pm0.001}$ | $0.315_{\pm0.098}$ | $0.018_{\pm0.012}$ | $0.014_{\pm0.008}$ |
| ✓ | ✗ | $89.3_{\pm3.6}$ | $\mathbf{0.000}_{\pm0.000}$ | $0.019_{\pm0.013}$ | $\underline{0.047}_{\pm0.077}$ | $0.005_{\pm0.011}$ | $\mathbf{0.169}_{\pm0.106}$ | $0.041_{\pm0.082}$ | $\underline{0.013}_{\pm0.010}$ |
| ✗ | ✗ | $\mathbf{95.6}_{\pm3.7}$ | $\mathbf{0.000}_{\pm0.000}$ | $0.042_{\pm0.042}$ | $0.077_{\pm0.091}$ | $0.005_{\pm0.003}$ | $0.217_{\pm0.169}$ | $\mathbf{0.013}_{\pm0.013}$ | $\mathbf{0.008}_{\pm0.011}$ |

| Node Budget | Minibatch OT | NumDiff ↓ | Deg ↓ | EdgeSize ↓ | Spectral ↓ | Harmonic ↓ | Closeness ↓ | Betweenness ↓ |
|---|---|---|---|---|---|---|---|---|
| | | | | **ModelNet Bookshelf** | | | | |
| ✓ | ✓ | $\mathbf{0.135}_{\pm0.276}$ | $\underline{2.980}_{\pm2.107}$ | $\mathbf{0.020}_{\pm0.025}$ | $\underline{0.024}_{\pm0.015}$ | $\mathbf{46.614}_{\pm58.803}$ | $0.086_{\pm0.030}$ | $\mathbf{0.001}_{\pm0.001}$ |
| ✗ | ✓ | $0.940_{\pm0.917}$ | $3.040_{\pm1.446}$ | $0.089_{\pm0.099}$ | $0.032_{\pm0.013}$ | $59.300_{\pm108.101}$ | $0.137_{\pm0.042}$ | $0.003_{\pm0.001}$ |
| ✓ | ✗ | $\underline{0.265}_{\pm0.496}$ | $4.400_{\pm1.635}$ | $0.025_{\pm0.021}$ | $\mathbf{0.014}_{\pm0.007}$ | $54.558_{\pm30.276}$ | $0.102_{\pm0.020}$ | $\mathbf{0.001}_{\pm0.001}$ |
| ✗ | ✗ | $1.325_{\pm1.631}$ | $\mathbf{2.820}_{\pm0.958}$ | $0.055_{\pm0.077}$ | $0.031_{\pm0.009}$ | $79.889_{\pm98.721}$ | $0.135_{\pm0.012}$ | $0.003_{\pm0.001}$ |
| | | | | **ModelNet Piano** | | | | |
| ✓ | ✓ | $\mathbf{0.846}_{\pm1.009}$ | $\mathbf{3.265}_{\pm1.954}$ | $\mathbf{0.042}_{\pm0.056}$ | $0.040_{\pm0.026}$ | $119.158_{\pm81.023}$ | $\mathbf{0.123}_{\pm0.119}$ | $\mathbf{0.002}_{\pm0.002}$ |
| ✗ | ✓ | $3.622_{\pm1.822}$ | $\underline{3.358}_{\pm1.772}$ | $\underline{0.054}_{\pm0.080}$ | $0.055_{\pm0.040}$ | $133.806_{\pm137.603}$ | $0.155_{\pm0.058}$ | $0.012_{\pm0.033}$ |
| ✓ | ✗ | $\underline{3.155}_{\pm3.637}$ | $5.250_{\pm1.087}$ | $0.066_{\pm0.168}$ | $\mathbf{0.030}_{\pm0.048}$ | $154.211_{\pm383.979}$ | $0.188_{\pm0.164}$ | $\mathbf{0.002}_{\pm0.002}$ |
| ✗ | ✗ | $5.490_{\pm8.847}$ | $3.733_{\pm2.989}$ | $0.107_{\pm0.215}$ | $\underline{0.036}_{\pm0.016}$ | $\mathbf{97.724}_{\pm198.774}$ | $\underline{0.138}_{\pm0.129}$ | $0.002_{\pm0.002}$ |

Table 12: Detailed numerical results for ablation studies on unfeatured hypergraphs.

| Node Budget | Minibatch OT | ChamDist ↓ | NumDiff ↓ | Deg ↓ | EdgeSize ↓ | Spectral ↓ | Harmonic ↓ | Closeness ↓ | Betweenness ↓ |
|---|---|---|---|---|---|---|---|---|---|
| | | | | **ManifoldNet Bench** | | | | | |
| ✓ | ✓ | $\mathbf{0.064}_{\pm0.005}$ | $\underline{0.060}_{\pm0.149}$ | $\underline{0.349}_{\pm0.454}$ | $\underline{0.009}_{\pm0.017}$ | $\underline{0.017}_{\pm0.011}$ | $5.069_{\pm8.171}$ | $0.046_{\pm0.040}$ | $\mathbf{0.004}_{\pm0.003}$ |
| ✗ | ✓ | $0.090_{\pm0.003}$ | $0.240_{\pm0.265}$ | $1.089_{\pm0.346}$ | $0.013_{\pm0.015}$ | $0.018_{\pm0.009}$ | $6.507_{\pm2.771}$ | $\mathbf{0.014}_{\pm0.005}$ | $\underline{0.005}_{\pm0.002}$ |
| ✓ | ✗ | $\underline{0.085}_{\pm0.056}$ | $\mathbf{0.020}_{\pm0.032}$ | $\mathbf{0.221}_{\pm0.160}$ | $\mathbf{0.006}_{\pm0.004}$ | $\mathbf{0.013}_{\pm0.003}$ | $\mathbf{1.864}_{\pm2.828}$ | $0.028_{\pm0.019}$ | $0.005_{\pm0.006}$ |
| ✗ | ✗ | $0.098_{\pm0.024}$ | $0.267_{\pm0.319}$ | $1.242_{\pm0.582}$ | $0.013_{\pm0.011}$ | $0.022_{\pm0.014}$ | $7.619_{\pm4.165}$ | $\underline{0.015}_{\pm0.016}$ | $0.006_{\pm0.004}$ |
| | | | | **ManifoldNet Airplane** | | | | | |
| ✓ | ✓ | $\mathbf{0.048}_{\pm0.003}$ | $\mathbf{0.017}_{\pm0.070}$ | $\underline{0.218}_{\pm0.085}$ | $\mathbf{0.004}_{\pm0.007}$ | $\underline{0.010}_{\pm0.003}$ | $\underline{2.428}_{\pm1.455}$ | $0.032_{\pm0.007}$ | $\mathbf{0.003}_{\pm0.001}$ |
| ✗ | ✓ | $0.079_{\pm0.019}$ | $0.426_{\pm0.820}$ | $0.588_{\pm0.451}$ | $0.024_{\pm0.021}$ | $0.014_{\pm0.007}$ | $2.429_{\pm1.394}$ | $0.033_{\pm0.028}$ | $\mathbf{0.003}_{\pm0.002}$ |
| ✓ | ✗ | $\underline{0.050}_{\pm0.005}$ | $\underline{0.052}_{\pm0.065}$ | $\mathbf{0.235}_{\pm0.063}$ | $\underline{0.014}_{\pm0.027}$ | $\mathbf{0.012}_{\pm0.005}$ | $\mathbf{1.604}_{\pm1.400}$ | $\mathbf{0.022}_{\pm0.007}$ | $\mathbf{0.003}_{\pm0.001}$ |
| ✗ | ✗ | $0.100_{\pm0.023}$ | $0.304_{\pm0.437}$ | $0.864_{\pm0.358}$ | $0.020_{\pm0.016}$ | $0.019_{\pm0.009}$ | $4.184_{\pm2.176}$ | $\underline{0.025}_{\pm0.009}$ | $0.005_{\pm0.002}$ |

Table 13: Detailed numerical results for ablation studies on 3D meshes.

# I  COMPARISON BETWEEN TRAINING AND GENERATED SAMPLES

Train samples          Generated samples

(*i*) Stochastic Block Model hypergraphs.

Train samples          Generated samples

(*ii*) Ego hypergraphs.

Train samples          Generated samples

(*iii*) Tree hypergraphs.

Train samples          Generated samples

(*iv*) Bookshelf meshes topology.

Train samples          Generated samples

(v) Piano meshes topology

Train samples          Generated samples

(*vi*) Stochastic Block Model graphs.

Train samples          Generated samples

(*vii*) Tree graphs.

Train samples          Generated samples

(*viii*) Planar graphs.

Train samples          Generated samples

(*ix*) Proteins.

Train samples          Generated samples

(*x*) Point clouds (only topology).

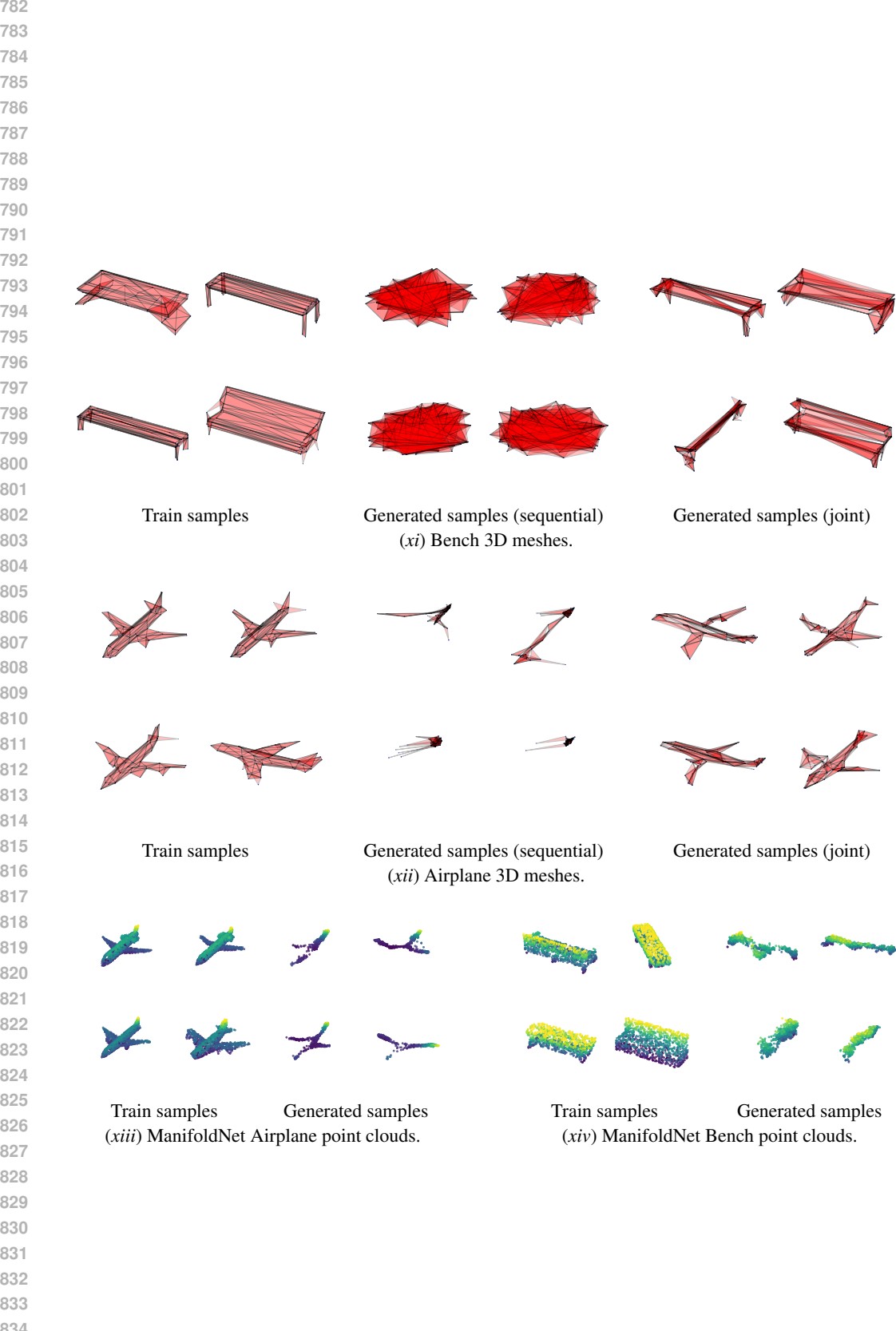

Train samples      Generated samples (sequential)      Generated samples (joint)

(*xi*) Bench 3D meshes.

Train samples      Generated samples (sequential)      Generated samples (joint)

(*xii*) Airplane 3D meshes.

Train samples      Generated samples          Train samples      Generated samples

(*xiii*) ManifoldNet Airplane point clouds.      (*xiv*) ManifoldNet Bench point clouds.

