# OpenReview forum: "Feature-aware (Hyper)graph Generation via Next-Scale Prediction"
_ICLR.cc/2026/Conference — Submitted to ICLR 2026_

### Official Review · Reviewer_VUAq · 2025-10-29

**Soundness:** 3
**Presentation:** 3
**Contribution:** 3
**Rating:** 6
**Confidence:** 3

**Summary:**

This paper studies the problem of hypergraph generation. The authors introduce FAHNES, a hierarchical framework that jointly generates hypergraph topology and features. FAHNES achieves state-of-the-art performance in several benchmarks.

**Strengths:**

- The paper is well-written and easy to follow
- The authors conduct ablation studies to analyze the effectiveness of the node-budget and OT-coupling components.

**Weaknesses:**

- Hierarchical graph generation models have been commonly applied in molecular generation. Several important works [1-3] should be included in the related work, and their difference should be discussed.

- In the experiments, the statistics of the used datasets should be provided to show the size of the dataset and the graph.

- As FAHNES is based on flow matching, it might be less efficient than one-shot methods. The inference time should be reported for a more comprehensive comparison between quality and cost.

- In Table 3, all the baselines are OOM. Hence, the performance of FAHNES cannot be compared.

[1] Coarse-to-fine: a hierarchical diffusion model for molecule generation in 3d

[2] MolGrow: A graph normalizing flow for hierarchical molecular generation

[3] Molhf: A hierarchical normalizing flow for molecular graph generation

**Questions:**

See weaknesses

---

> ### Author Response · Authors · 2025-11-20
> **Official Comment by Authors (part 1)**
>
> **Weaknesses:**
>
> **W1.** We have updated the *Related work* section of our paper. Regarding the differences, [1] uses a two-step process where first “high-level” group/cluster representations are generated, before a second pass determines the type of the cluster/functional group and the connectivity. Extending this framework to more than two hierarchical steps is highly non-trivial, and all the steps of the algorithm are heavily tailored for molecular generation. Our method instead determines all features and connectivity jointly at every scale, and our algorithm is scale-invariant, easily allowing for stacking as many hierarchical steps as needed.
>
> [2] and [3] do not use teacher forcing to train the different steps of generation, and instead rely on a complicated dequantization technique to allow backpropagation through discrete data across scales. They use a breadth-first search algorithm (or a variation of it adapted to molecules) to order nodes for merging during coarsening, whereas we use the much more flexible and robust local variation cost introduced in [4]. Additionally, they don’t merge the features but concatenate, which restricts the possible choices in architecture, as it needs to be permutation invariant.
>
> Finally, all of these methods are tailored for molecule generation, whereas ours is more general and scalable.
>
> **W2.** We have revised the manuscript to make the statistics of the datasets more evident. These statistics were already present in Appendix D (lines 903-918, 959-964, and 982-984).
>
> **W3.** We already discussed the complexity in Appendix G, which is linear in the number of nodes, hyperedges, and incidences, and is also linear in the number of steps during flow-matching.
>
> We stress that flow-matching is not central to our method and can be replaced by any off-the-shelf generative method like GANs or VAEs (where OT-coupling can still be used to solve the permutation invariance problem). As all modern graph generation methods rely on diffusion/flow matching, we only tested flow-matching. The complexity of one step of flow-matching is the same as the prediction complexity of GANs or VAEs; thus, the only difference comes from the number of steps.
>
> Thus, the important comparison would be between hierarchical methods and one-shot methods. Such a comparison has already been done in [5] (see their Appendix H, Figure 6), showing that hierarchical methods have an approximately logarithmic sampling time complexity against a quadratic time complexity for flat methods. Flat methods are thus faster for small graphs but lose their advantage for large graphs.
>
> **W4.** We’ve added experiments on the five small-to-large graph datasets used by [5] using the proposed graph adaptation of our method in Section 3.6. We added those to the paper (Tables 3 and 4) and summarized the results in the following tables:
>
> ### SBM graphs (n_avg = 105.99, std = 38.38)
>
> | Method   | Valid SBM ↑      | Spectral ↓         | Ratio ↓          |
> |----------|------------------|---------------------|-------------------|
> | HSpectre | 45.0             | 0.007               | 10.2              |
> | BwR      | 7.5              | 0.017               | 38.6              |
> | DiGress  | _60.0_           | **0.004**           | **1.7**           |
> | DeFoG    | **90.0 ± 5.1**   | 0.005 ± 0.001       | 4.9 ± 1.3         |
> | **FAHNES** | 50.0 ± 5.0       | **0.004 ± 0.002**   | _4.8 ± 0.7_       |
>
> ---
>
> ### Tree graphs (n_avg = 64.0, std = 0)
>
> | Method   | Valid Tree ↑     | Spectral ↓         | Ratio ↓          |
> |----------|------------------|---------------------|-------------------|
> | HSpectre | **100.0**        | 0.012               | 4.0               |
> | BwR      | 0.0              | 0.048               | 11.4              |
> | DiGress  | 90.0             | **0.011**           | **1.6**           |
> | DeFoG    | 96.5 ± 2.6       | **0.011 ± 0.002**   | **1.6 ± 0.4**     |
> | **FAHNES** | **100.0 ± 0.0** | 0.012 ± 0.004       | _1.8 ± 0.8_       |
>
> ---
>
> ### Planar graphs (n_avg = 64.0, std = 0.0)
>
> | Method   | Valid Planar ↑   | Spectral ↓         | Ratio ↓          |
> |----------|------------------|---------------------|-------------------|
> | HSpectre | 95.0             | _0.008_             | _2.1_             |
> | BwR      | 0.0              | 0.044               | 251.9             |
> | DiGress  | 77.5             | 0.010               | 5.1               |
> | DeFoG    | **99.5 ± 1.0**   | **0.007 ± 0.001**   | **1.6 ± 0.04**    |
> | **FAHNES** | _96.7 ± 6.2_     | _0.008 ± 0.003_     | 2.2 ± 1.6         |
>
> ---
>
> ### Protein graphs (n_avg = 261.54, std = 104.68)
>
> | Method   | Spectral ↓         | Ratio ↓          |
> |----------|---------------------|-------------------|
> | HSpectre | **0.001**          | _5.9_             |
> | BwR      | 0.070              | 254.4             |
> | DiGress  | 0.002              | 18.0              |
> | DeFoG    | —                  | —                 |
> | **FAHNES** | **0.001 ± 0.001** | **3.8 ± 1.9**     |

---

> ### Author Response · Authors · 2025-11-20
> **Official Comment by Authors (part 2)**
>
> ### PointCloud graphs (n_avg = 1434.31, std = 1285.93)
>
> | Method   | Spectral ↓         | Ratio ↓          |
> |----------|---------------------|-------------------|
> | HSpectre | **0.005**          | _7.0_             |
> | BwR      | 0.291              | 133.2             |
> | DiGress  | OOM                | OOM               |
> | DeFoG    | —                  | —                 |
> | **FAHNES** | **0.005 ± 0.001** | **3.2 ± 0.5**     |
>
> Detailed results can be found in Table 10 of Appendix H in the revised version of the paper. This shows that our method remains competitive for small- and medium-scale graph generation, while enjoying a large scalability advantage. This also shows that our additional components improve on the hierarchical method, HSpectre, introduced by [5].
>
> [4] Loukas, A. (2019). Graph reduction with spectral and cut guarantees. Journal of Machine Learning Research
>
> [5] Bergmeister et al., "Efficient and Scalable Graph Generation," ICLR 2024

---

> ### Author Response · Authors · 2025-11-27
> **Follow-Up on Rebuttal**
>
> We thank the reviewer again for the thoughtful and constructive comments, which have been helpful for improving our submission.
>
> As the author–reviewer discussion period is coming to an end, we would like to kindly confirm that the reviewer have had a chance to review our rebuttal. We believe that the main concerns raised are now thoroughly addressed in our response and revisions.
>
> Additionally, because these updates and clarifications were made specifically in response to the constructive feedback, we kindly hope that they will encourage the reviewer to reconsider the evaluation. Should there be any remaining questions or reservations, please feel free to let us know at any time. We are fully committed to addressing all concerns.

---

### Official Review · Reviewer_mWok · 2025-10-30

**Soundness:** 3
**Presentation:** 3
**Contribution:** 3
**Rating:** 6
**Confidence:** 2

**Summary:**

The paper introduces FAHNES for joint generative modeling of topology and features on hypergraphs (and graphs). It learns to invert a multi-scale coarsening to expansion process: standard spectral coarsening is applied on the clique expansion to obtain coarse levels; generation then happens on the bipartite (star) expansion via a flow-matching model that expands clusters and refines features from coarse to fine scales. A node-budget mechanism controls local growth and final size, and minibatch OT-coupling aligns permutations to stabilize training. Experiments on synthetic hypergraphs, 3D meshes, and point clouds show strong fidelity while scaling beyond flat/disjoint baselines.

**Strengths:**

1. The paper addresses the notorious permutation misalignment by constrained OT-coupling; the inclusion of an algorithm box and restricted local permutations (2 or 6) in each cluster shows care for both correctness and efficiency.
2. Many application domains (e.g., 3D geometry, circuits) need features as much as topology. Moving hierarchical generation into the feature-aware hypergraph regime is impactful for practitioners who cannot rely on topology-only generators.
3. By replacing flat, quadratic-cost modeling with a multi-scale next-scale prediction approach, the paper provides a credible path to larger instances without discarding features, a key limitation of many prior methods.

**Weaknesses:**

1. The 3D mesh setup appears two-stage: first learn topology, then generate coordinates with a separate Local-PPGN flow-matching model. It’s unclear whether FAHNES truly jointly models topology and features in this domain or only the topology (with features predicted post-hoc).
2. The refinement step uses fixed thresholds (e.g., 0.5 for edges). No analysis of calibration or threshold selection is provided.
3. OT-coupling is only applied within the siblings created by a single cluster expansion (2 or 6 permutations), with strict equivalence constraints on structure, budgets, and features. While elegant, this restriction may limit alignment benefits, and the ablation gains look small/uneven across datasets.

**Questions:**

1. Coarsening is performed on the clique expansion (Loukas) while expansion/refinement are learned on the bipartite representation. Please discuss the trade-offs (spectral preservation vs. hyperedge degree distortion) and whether you observed systematic topology artifacts when mapping between the two domains. Any guarantees that hyperedge cardinalities are preserved in expectation?
2. others see weakness

---

> ### Author Response · Authors · 2025-11-20
> **Official Comment by Authors**
>
> **Weaknesses:**
>
> **W1.** FAHNES uses the same framework for features as for topology and does joint predictions for both at every level in a hierarchical manner, please see Eqn. (7). The setup described by the reviewer is the sequential two-stage baseline, not our proposed method.
>
> **W2.** As the model is trained to output discrete values (-1 and 1 for nodes and incidences, and -1, 0, and 1 for hyperedges), this thresholding step is simply a projection of the continuous values predicted by the model onto the real possible values. This can be seen as a rounding, where calibration is not needed.
>
> **W3.** Permutating nodes for alignment is extremely costly (for example, the Hungarian algorithm has a cubic complexity in the number of nodes), whereas strictly limiting it to children of expanded clusters makes its additional overhead negligible. While the ablation gains are small across simple datasets, they are much more pronounced for the complex features of 3D meshes, particularly for the Airplane dataset, as shown in Table 7.
>
> **Questions:**
>
> **Q1.** During coarsening, we have equivalence between the clique expansion and the bipartite representation, as every node merging in the clique expansion is mirrored in the bipartite representation. As hyperedge cardinality is central to the computation of the hypergraph Laplacian, we do not see an immediate tradeoff between preserving the spectrum and the hyperedge cardinality. Besides, hyperedge cardinalities cannot be preserved in expectation as the number of nodes necessarily decreases during coarsening, until a single node remains.

---

> > ### Comment · Reviewer_mWok · 2025-11-25
> >
> > Thanks authors for the rebuttal, I will keep my accept score.

---

### Official Review · Reviewer_qU8D · 2025-10-30

**Soundness:** 2
**Presentation:** 2
**Contribution:** 3
**Rating:** 4
**Confidence:** 3

**Summary:**

This paper proposes FAHNES, a hierarchical generative model for (hyper)graphs.  It extends the recent hierarchical graph generator [1] and hypergraph generator (HYGENE [2]) by introducing a node budget mechanism and minibatch optimal transport (OT) coupling to align flow-matching predictions across scales.  The model jointly generates topology and features through a flow-matching ODE trained across multiple coarsening levels. Experiments on synthetic, 3D mesh, and point-cloud datasets show promising results compared to diffusion- and VAE-based baselines.

**Strengths:**

- The paper is very well-written and full of details.

- The authors conduct comprehensive experiments.

- The ablations are thorough, and the complexity discussion provides useful insight into scalability.

**Weaknesses:**

1. The main ideas—representing hypergraphs as bipartite graphs and using a coarsening–refinement hierarchy—are largely borrowed from [1] and HYGENE [2]. FAHNES mainly replaces the diffusion-based training paradigm with flow matching and adds the budget and OT-coupling components, which are meaningful engineering refinements but not a fundamental conceptual breakthrough.

2. The model depends on a complete bipartite representation of hypergraphs, which is computationally expensive and scales poorly.

3. The model heavily relies on Laplacian spectral features and SignNet encodings, which require repeated eigen-decompositions across multiple levels. This design becomes computationally challenging and limits scalability as the graph or hypergraph size increases.

4. The core flow-matching objective is described in Appendix E.  The main paper should include a clear formulation of the training loss and explicitly connect it to the main model variables (v, e, f, F)

5. Limited and partially unfair baselines. On the graph point cloud datasets, the paper only compares FAHNES to DiGress and DeFoG, both of which are designed for discrete graph generation rather than point clouds. Reporting “OOM” results for these baselines is not informative or fair. Similarly, for 3D mesh generation, only very limited baselines are considered; comparison with more relevant mesh or point-cloud generation methods would make the evaluation stronger.

6. The paper does not mention a code release, nor does it provide sufficient implementation details (e.g., architecture parameters, training hyperparameters) to ensure reproducibility.

[1] Efficient and Scalable Graph Generation through Iterative Local Expansion (ICLR, 2024)

[2] HYGENE: A Diffusion-based Hypergraph Generation Method (AAAI, 2025)

**Questions:**

1. Why use Spectrum-preserving coarsening in the node pair merging process?

2. Beyond introducing the budget and OT-coupling mechanisms, how does FAHNES fundamentally differ from HYGENE [2] in methodology?

3. During generation (from level L → L−1 → L−2 → … → 1), does each level require predicting a separate velocity field for flow matching, or is the same neural network shared across all levels?

---

> ### Author Response · Authors · 2025-11-20
> **Answers to Weaknesses 1-3**
>
> **Weaknesses:**
>
> **W1.** [1] and [2] are unable to generate features; our work is the first hierarchical method to enable the generation of features at a large scale. Our ablation studies demonstrate that our two components are essential for the generation of features, as shown, for example, by the large worsening in Chamfer Distance when ablating one of them—the Airplane and Bench 3D mesh datasets go from respectively **0.048** and **0.064** to **0.100** and **0.098** (108% and 53% increase). This metric has a logarithmic relation with the level of quality.
>
> For these reasons, we do not believe our contributions can be characterized as “engineering refinements”. If the reviewer could elaborate on the meaning of this term, we could answer objectively to this weakness.
>
> **W2.** We would like to clarify that the bipartite representation is not computationally prohibitive. Denoting $A$ the adjacency matrix of the bipartite representation and $B$ the incidence matrix of the hypergraph, we have:
>
>
> $$A_{\mathrm{bip}} =$$
>
>  \begin{bmatrix} 0_{n\times n} & B \end{bmatrix}
>
>  \begin{bmatrix} B^\top & 0_{m\times m} \end{bmatrix}
>
> Where $0_{n\times n}$ is a matrix of zeros of dimension $n \times n$. Therefore, switching from the hypergraph to the bipartite representation is almost instantaneous. Similarly, it is not true that it scales poorly, as its memory complexity is linear in the number of incidences ($\mathcal O (k)$, where $k$ is the number of edges in the bipartite representation), *i.e.*, the sum of the sizes of all hyperedges.
>
> **W3.** For SignNet, we do not use the full spectrum but only a fixed number $K$ of them, where $K$ is typically much smaller than the size of the graph. As written in Appendix G, using the algorithm by [3], the complexity is $\mathcal{O}(n + m + k)$, where $n$ is the number of nodes, $m$ the number of hyperedges, and $k$ the number of incidences, which is linear. Furthermore, the model does not strictly require SignNet and even performs better without it in some cases. For example, our experiments on 3D meshes and point clouds do not compute the eigenvalues and eigenvectors of the graph and do not rely on them (see the updated Appendix D.2 and D.3). We have updated Appendix D to explicitly state that.
>
> For spectrum-preserving coarsening, the same algorithm by [3] is used to compute the top $L$ eigenvalues and eigenvectors (we use $L = 8$ in our experiments). Therefore, the cost is linear in the number of nodes, hyperedges, and incidences, and does not prohibit scaling. Besides, the spectrum-preserving coarsening could be replaced by another selection based on feature similarity. For example, using a heuristic based on the distance between features of adjacent nodes, we can have competitive results as shown in the following table:
>
> ### Manifold40 Bench
>
> | Method              | Cham Dist ↓ | Node Num ↓ | Node Deg ↓ | Edge Size ↓ | Spectral ↓ |
> |---------------------|-------------|------------|-------------|--------------|-------------|
> | Spectrum-Preserving | 0.073       | 0.067      | 0.581       | 0.008        | 0.014       |
> | Feature-Preserving  | 0.085       | 0.200      | 0.750       | 0.004        | 0.030       |
>
> ---
>
> ### Manifold40 Airplane
>
> | Method              | Cham Dist ↓ | Node Num ↓ | Node Deg ↓ | Edge Size ↓ | Spectral ↓ |
> |---------------------|-------------|------------|-------------|--------------|-------------|
> | Spectrum-Preserving | 0.049       | 0          | 0.304       | 0.033        | 0.015       |
> | Feature-Preserving  | 0.056       | 0          | 0.241       | 0.002        | 0.014       |
>
>
> [3] Vishnoi, N. K. (2013). *Lx = b*. **Foundations and Trends in Theoretical Computer Science**, 8(1–2), 1–141.

---

> ### Author Response · Authors · 2025-11-20
> **Answers to weaknesses 4-5**
>
> **W4.** Since our method uses graph inpainting and filters out nodes during loss computation, the full loss explanation is lengthy, so we moved it to Appendix E and added the following paragraph.
>
> In classical flow-matching, the model is trained to minimize the expected squared error between the predicted and true endpoints:
> \begin{equation}
>     \mathcal{L} = \mathbb{E}_{t, \: \mathbf{x}_0 \sim p_0, \: \mathbf{x}_1 \sim p_1}
>     \left[ \left\| \hat{\mathbf{x}}_1(\mathbf{x}_t, 1) - \mathbf{x}_1 \right\|^2 \right],
> \end{equation}
> where $\mathbf{x}_t = (1 - t) \mathbf{x}_0 + t \mathbf{x}_1$ is a linear interpolation between samples from the prior and target distributions.
> Following \cite{esser2024scalingrectifiedflowtransformers}, during training, $t$ is sampled using a log-normal distribution, which has been shown to improve performance.
>
> In our setting, each loss term is masked so that it is evaluated only on the relevant nodes or hyperedges.
> Specifically, we retain:
> *(i)* expansion losses only for nodes with budget greater than one ;
> *(ii)* budget losses only for expanded nodes whose parent has a budget greater than two ; and
> *(iii)* feature losses only for expanded nodes.
> For a given sample, let
>
> $$
> \mathcal{L}_{\text{node-expansion}}
> = \left\| \hat{\mathbf{v}}_L - \mathbf{v}_L \right\|^2 \; \mathbf{1}[\mathbf{b} > 1],
> $$
>
> $$
> \mathcal{L}_{\text{budget}}
> = \left\| \hat{\mathbf{f}} - \mathbf{f} \right\|^2 \; \mathbf{1}[\mathbf{b} > 2 \wedge \text{child of expanded cluster}],
> $$
>
> $$
> \mathcal{L}_{\text{node-feature}}
> = \left\| \hat{\mathbf{F}}_L - \mathbf{F}_L^{\text{refine}} \right\|^2 \; \mathbf{1}[\text{child of expanded cluster}],
> $$
>
> $$
> \mathcal{L}_{\text{hyperedge-expansion}}
> = \left\| \hat{\mathbf{v}}_R - \mathbf{v}_R \right\|^2,
> $$
>
> $$
> \mathcal{L}_{\text{hyperedge-feature}}
> = \left\| \hat{\mathbf{F}}_R - \mathbf{F}_R^{\text{refine}} \right\|^2 \; \mathbf{1}[\text{child of expanded cluster}],
> $$
>
> $$
> \mathcal{L}_{\text{incidence}}
> = \left\| \hat{\mathbf{e}} - \mathbf{e} \right\|^2.
> $$
>
> where $\mathbf{1}[\cdot]$ denotes the indicator function.
> The total loss is the sum of all components, averaged across samples.
>
> **W5.** We’ve added experiments on the five small-to-large graph datasets used by [1] using the proposed graph adaptation of our method in Section 3.6. We added those to the paper (Tables 3 and 4) and summarized the results in the following tables:
>
> ### SBM graphs (n_avg = 105.99, std = 38.38)
>
> | Method   | Valid SBM ↑      | Spectral ↓         | Ratio ↓          |
> |----------|------------------|---------------------|-------------------|
> | HSpectre | 45.0             | 0.007               | 10.2              |
> | BwR      | 7.5              | 0.017               | 38.6              |
> | DiGress  | _60.0_           | **0.004**           | **1.7**           |
> | DeFoG    | **90.0 ± 5.1**   | 0.005 ± 0.001       | 4.9 ± 1.3         |
> | **FAHNES** | 50.0 ± 5.0       | **0.004 ± 0.002**   | _4.8 ± 0.7_       |
>
> ---
>
> ### Tree graphs (n_avg = 64.0, std = 0)
>
> | Method   | Valid Tree ↑     | Spectral ↓         | Ratio ↓          |
> |----------|------------------|---------------------|-------------------|
> | HSpectre | **100.0**        | 0.012               | 4.0               |
> | BwR      | 0.0              | 0.048               | 11.4              |
> | DiGress  | 90.0             | **0.011**           | **1.6**           |
> | DeFoG    | 96.5 ± 2.6       | **0.011 ± 0.002**   | **1.6 ± 0.4**     |
> | **FAHNES** | **100.0 ± 0.0** | 0.012 ± 0.004       | _1.8 ± 0.8_       |
>
> ---
>
> ### Planar graphs (n_avg = 64.0, std = 0.0)
>
> | Method   | Valid Planar ↑   | Spectral ↓         | Ratio ↓          |
> |----------|------------------|---------------------|-------------------|
> | HSpectre | 95.0             | _0.008_             | _2.1_             |
> | BwR      | 0.0              | 0.044               | 251.9             |
> | DiGress  | 77.5             | 0.010               | 5.1               |
> | DeFoG    | **99.5 ± 1.0**   | **0.007 ± 0.001**   | **1.6 ± 0.04**    |
> | **FAHNES** | _96.7 ± 6.2_     | _0.008 ± 0.003_     | 2.2 ± 1.6         |
>
> ---
>
> ### Protein graphs (n_avg = 261.54, std = 104.68)
>
> | Method   | Spectral ↓         | Ratio ↓          |
> |----------|---------------------|-------------------|
> | HSpectre | **0.001**          | _5.9_             |
> | BwR      | 0.070              | 254.4             |
> | DiGress  | 0.002              | 18.0              |
> | DeFoG    | —                  | —                 |
> | **FAHNES** | **0.001 ± 0.001** | **3.8 ± 1.9**     |
>
> ---
>
> ### PointCloud graphs (n_avg = 1434.31, std = 1285.93)
>
> | Method   | Spectral ↓         | Ratio ↓          |
> |----------|---------------------|-------------------|
> | HSpectre | **0.005**          | _7.0_             |
> | BwR      | 0.291              | 133.2             |
> | DiGress  | OOM                | OOM               |
> | DeFoG    | —                  | —                 |
> | **FAHNES** | **0.005 ± 0.001** | **3.2 ± 0.5**     |

---

> ### Author Response · Authors · 2025-11-20
> **Answers to weaknesses 5-6 and Questions**
>
> Detailed results can be found in Table 10 of Appendix H in the revised version of the paper. This shows that our method remains competitive for small- and medium-scale graph generation, while enjoying a large scalability advantage. This also shows that our additional components improve on the hierarchical method, HSpectre, introduced by [1].
>
> Regarding graph-based point-cloud generation, we are not aware of any other relevant baselines beyond what we already included. For hypergraph-based mesh generation, to our knowledge, FAHNES is the first method that generates meshes based on hypergraphs. Typical methodologies treat mesh generation as a sequential, autoregressive decoding problem that tokenizes meshes and uses transformer-based models to predict those tokens step-by-step, which is not relevant in our case, as our goal is benchmarking graph generation, not targeting a specific modality. If the reviewer could suggest specific graph- or hypergraph-based generative models relevant to our work, we could add the baseline.
>
>
> **W6.** Our reproducibility statement states that the code will be made public upon acceptance of the paper. Besides, Appendices B, D, and E provide architectural details and hyperparameters.
>
> **Questions:**
>
> **Q1.** [1, 2] already ablated on the spectrum-preserving coarsening and found better results with it. We have done this experiment again on our 3D mesh dataset as shown in the following table:
>
> ### Manifold40 Bench
>
> | Method              | Cham Dist ↓ | Node Num ↓ | Node Deg ↓ | Edge Size ↓ | Spectral ↓ |
> |---------------------|------------|------------|------------|-------------|------------|
> | Spectrum-Preserving | 0.073      | 0.067      | 0.581      | 0.008       | 0.014      |
> | Random Mergings     | 0.158      | 2.300      | 0.324      | 0.002       | 0.019      |
>
> ### Manifold40 Airplane
>
> | Method              | Cham Dist ↓ | Node Num ↓ | Node Deg ↓ | Edge Size ↓ | Spectral ↓ |
> |---------------------|------------|------------|------------|-------------|------------|
> | Spectrum-Preserving | 0.049      | 0          | 0.304      | 0.033       | 0.015      |
> | Random Mergings     | 0.054      | 0          | 0.423      | 0.009       | 0.011      |
>
> We have the same takeaways from [1, 2], namely that spectrum-preserving coarsening improves generation quality.
>
> **Q2.** (i) HYGENE is unable to generate features. (ii) Please note that graph equivalence by permutation is a major, open problem in graph generation (*e.g.*, see section 3.4 in [3]), which often requires heuristics or computationally expensive algorithms. Here, we elegantly solved this problem with a negligible additional cost in our setting with OT-coupling.
>
> **Q3.** The same model is shared across all scales. For every level, the model outputs a prediction for every node, hyperedge, and incidence, which requires $N$ steps of flow-matching, *i.e.* $N$ velocity field predictions (in our experiments, we use $N = 256$).
>
> [3] Simonovsky, M., & Komodakis, N. (2018, September). Graphvae: Towards generation of small graphs using variational autoencoders. International Conference on Artificial Neural Networks

---

> ### Author Response · Authors · 2025-11-27
> **Follow-Up on Rebuttal**
>
> We thank the reviewer again for the thoughtful and constructive comments, which have been helpful for improving our submission.
>
> As the author–reviewer discussion period is coming to an end, we would like to kindly confirm that the reviewer have had a chance to review our rebuttal. We believe that the main concerns raised are now thoroughly addressed in our response and revisions.
>
> Additionally, because these updates and clarifications were made specifically in response to the constructive feedback, we kindly hope that they will encourage the reviewer to reconsider the evaluation. Should there be any remaining questions or reservations, please feel free to let us know at any time. We are fully committed to addressing all concerns.

---

### Official Review · Reviewer_pQxt · 2025-11-10

**Soundness:** 3
**Presentation:** 2
**Contribution:** 3
**Rating:** 4
**Confidence:** 3

**Summary:**

This paper introduces FAHNES, a feature-aware hierarchical (hyper)graph generative framework that jointly models topology and features across multiple scales. The key idea is to perform next-scale prediction via a node budget mechanism that controls local graph growth and ensures cross-scale consistency. Additionally, the authors extend flow-matching optimal transport coupling (OT-coupling) to hierarchical structures to stabilize training and align node-level correspondences.

**Strengths:**

This paper introduces a feature-aware mechanism into a hierarchical generative framework, jointly modeling topology and node/hyperedge features.

Treating graphs as special cases of hypergraphs makes the approach theoretically general and practically versatile.

The paper evaluates FAHNES on diverse datasets, including both featureless and featured hypergraphs. Results demonstrate SOTA performance in topology–feature joint generation.

**Weaknesses:**

Current experiments only involve continuous geometric features (3D coordinates). The applicability to categorical or mixed-type features (e.g., molecular attributes) remains untested.

Although DiGress and DeFoG encounter OOM issues on our large-scale point cloud datasets, this does not necessarily imply that these models are inferior. They are primarily designed for small- and medium-scale discrete graphs, focusing on structural fidelity rather than scalability. A fair benchmarking protocol should therefore distinguish small-scale fidelity benchmarks (e.g., molecular graphs) from large-scale scalability benchmarks (e.g., 3D meshes or hypergraphs).

**Questions:**

How is error accumulation across scales mitigated in the node budget mechanism? Since the node budget mechanism recursively propagates predicted budget values through multiple scales, small prediction errors at higher levels could potentially amplify during expansion. Have the authors analyzed the stability or robustness of this mechanism to such accumulated errors?

The paper shows impressive results on hypergraphs, but in standard graph domains, evaluation seems insufficient. Moreover, the point cloud experiment does not allow a fair comparison with baseline models due to OOM errors. Could the authors clarify whether FAHNES has a measurable advantage on typical graphs such as molecules or social networks?

---

> ### Author Response · Authors · 2025-11-20
> **Official Comment by Authors**
>
> **Weaknesses:**
>
> **W1.** Adapting discrete data to our setting, beyond a naive one-hot encoding in continuous space, is non-trivial due to the hierarchical nature of our pipeline, which requires merging features at each scale. Thus, we cannot apply standard discrete methods out of the box. Designing a hierarchical mechanism for discrete features is an interesting direction for future work, but it falls outside the current scope.
>
> **W2.** We’ve added experiments on the five small-to-large graph datasets used by [1] using the proposed graph adaptation of our method in Section 3.6. We added those to the paper (Tables 3 and 4) and summarized the results in the following tables:
>
> ### SBM graphs (n_avg = 105.99, std = 38.38)
>
> | Method   | Valid SBM ↑      | Spectral ↓         | Ratio ↓          |
> |----------|------------------|---------------------|-------------------|
> | HSpectre | 45.0             | 0.007               | 10.2              |
> | BwR      | 7.5              | 0.017               | 38.6              |
> | DiGress  | _60.0_           | **0.004**           | **1.7**           |
> | DeFoG    | **90.0 ± 5.1**   | 0.005 ± 0.001       | 4.9 ± 1.3         |
> | **FAHNES** | 50.0 ± 5.0       | **0.004 ± 0.002**   | _4.8 ± 0.7_       |
>
> ---
>
> ### Tree graphs (n_avg = 64.0, std = 0)
>
> | Method   | Valid Tree ↑     | Spectral ↓         | Ratio ↓          |
> |----------|------------------|---------------------|-------------------|
> | HSpectre | **100.0**        | 0.012               | 4.0               |
> | BwR      | 0.0              | 0.048               | 11.4              |
> | DiGress  | 90.0             | **0.011**           | **1.6**           |
> | DeFoG    | 96.5 ± 2.6       | **0.011 ± 0.002**   | **1.6 ± 0.4**     |
> | **FAHNES** | **100.0 ± 0.0** | 0.012 ± 0.004       | _1.8 ± 0.8_       |
>
> ---
>
> ### Planar graphs (n_avg = 64.0, std = 0.0)
>
> | Method   | Valid Planar ↑   | Spectral ↓         | Ratio ↓          |
> |----------|------------------|---------------------|-------------------|
> | HSpectre | 95.0             | _0.008_             | _2.1_             |
> | BwR      | 0.0              | 0.044               | 251.9             |
> | DiGress  | 77.5             | 0.010               | 5.1               |
> | DeFoG    | **99.5 ± 1.0**   | **0.007 ± 0.001**   | **1.6 ± 0.04**    |
> | **FAHNES** | _96.7 ± 6.2_     | _0.008 ± 0.003_     | 2.2 ± 1.6         |
>
> ---
>
> ### Protein graphs (n_avg = 261.54, std = 104.68)
>
> | Method   | Spectral ↓         | Ratio ↓          |
> |----------|---------------------|-------------------|
> | HSpectre | **0.001**          | _5.9_             |
> | BwR      | 0.070              | 254.4             |
> | DiGress  | 0.002              | 18.0              |
> | DeFoG    | —                  | —                 |
> | **FAHNES** | **0.001 ± 0.001** | **3.8 ± 1.9**     |
>
> ---
>
> ### PointCloud graphs (n_avg = 1434.31, std = 1285.93)
>
> | Method   | Spectral ↓         | Ratio ↓          |
> |----------|---------------------|-------------------|
> | HSpectre | **0.005**          | _7.0_             |
> | BwR      | 0.291              | 133.2             |
> | DiGress  | OOM                | OOM               |
> | DeFoG    | —                  | —                 |
> | **FAHNES** | **0.005 ± 0.001** | **3.2 ± 0.5**     |
>
> Detailed results can be found in Table 10 of Appendix H in the revised version of the paper. This shows that our method remains competitive for small- and medium-scale graph generation, while enjoying a large scalability advantage. This also shows that our additional components improve on the hierarchical method, HSpectre, introduced by [1].
>
>
> **Questions:**
>
> **Q1.** Error accumulation is inherent to every autoregressive model. In our setting, budgets are integers, and the sum of the budgets of the children necessarily equals that of the parent node. Thus, during expansion, we project the two predicted budgets to the nearest integers satisfying this condition, using a Von-Neumann projection. See Appendix E.1 for the projection and line 277 for the rounding. Due to the increasing granularity of roundings - as the scale increases, budgets necessarily become smaller, allowing fewer possibilities when discretizing - thus the prediction errors become less and less important as generation progresses. As the starting point is always the same (a single node), the model strongly focuses on those first steps, mitigating the severity of error accumulations.
>
> **Q2.** We added results for regular graph generation datasets (see Weakness 2). FAHNES shows state-of-the-art or competitive results on small-to-large graph datasets, and enjoys a large scalability advantage over state-of-the-art flat (e.g., DeFoG, DiGress) methods.
>
> [1] Bergmeister et al., "Efficient and Scalable Graph Generation," ICLR 2024.

---

> ### Author Response · Authors · 2025-11-27
> **Follow-Up on Rebuttal**
>
> We thank the reviewer again for the thoughtful and constructive comments, which have been helpful for improving our submission.
>
> As the author–reviewer discussion period is coming to an end, we would like to kindly confirm that the reviewer have had a chance to review our rebuttal. We believe that the main concerns raised are now thoroughly addressed in our response and revisions.
>
> Additionally, because these updates and clarifications were made specifically in response to the constructive feedback, we kindly hope that they will encourage the reviewer to reconsider the evaluation. Should there be any remaining questions or reservations, please feel free to let us know at any time. We are fully committed to addressing all concerns.

---

### Meta-Review · Area_Chair_qVuu · 2025-12-27

**Summary:**

This paper proposes a hierarchical flow-matching framework for generating hypergraph topology and continuous features across scales via a node-budget expansion mechanism and minibatch OT-coupling to mitigate permutation misalignment. Reviewers agree the paper is clearly written and the goal (scalable featured hypergraph generation) is important. However, there are central concerns around how much conceptual novelty is beyond prior hierarchical graph/hypergraph generation, and fairness and strength of the empirical evaluation on standard graph-generation benchmarks. Reviewers also expressed concerns around the computational burden and reliance on design choices. The rebuttal adds additional graph benchmarks and technical clarifications, but the evidence remains insufficient to establish a clear, robust advantage over strong baselines on conventional graph domains.

**Reviewer Concerns:**

Partly addressed:
- Added extra graph benchmarks.
- Clarified joint topology+feature modeling and provided more implementation details.
- Gave arguments about bipartite + spectral cost/scaling.
- Expanded related work.

Still outstanding:
- Novelty concern: much of the hierarchy and bipartite setup follows prior work. The proposed node-budget and OT-coupling seem incremental.
- Evaluation not fully convincing: results are often only competitive on standard graph tasks. Limited relevant baselines for meshes/point clouds/hypergraphs.
- Efficiency unclear: no clean end-to-end runtime and inference comparison across scales.
- Limited feature scope: mainly continuous geometric features; categorical/mixed features deferred.

**Reviewer Scores:**

mWok: 6 and likely unchanged.

VUAq: 6 and likely unchanged.

pQxt: 4 and likely unchanged. Raised fairness of comparisons and feature-type scope; authors added benchmarks but key concerns likely remain.

qU8D: 4 and likely unchanged. Questioned conceptual novelty, baseline fairness, and computational overhead; rebuttal clarifies but may not overturn these points.

---

### Decision · Program_Chairs · 2026-01-26

Reject